# Optogenetic relaxation of actomyosin contractility uncovers mechanistic roles of cortical tension during cytokinesis

Kei Yamamoto [1,2,3], Haruko Miura[1,2], Motohiko Ishida[4,5], Yusuke Mii [1,2,3,6], Noriyuki Kinoshita[1,3], Shinji Takada [1,2,3], Naoto Ueno[1,3,7], Satoshi Sawai [4,5], Yohei Kondo[1,2,3,8 ✉] & Kazuhiro Aoki [1,2,3,7,8 ✉]

Actomyosin contractility generated cooperatively by nonmuscle myosin II and actin filaments plays essential roles in a wide range of biological processes, such as cell motility, cytokinesis, and tissue morphogenesis. However, subcellular dynamics of actomyosin contractility underlying such processes remains elusive. Here, we demonstrate an optogenetic method to induce relaxation of actomyosin contractility at the subcellular level. The system, named OptoMYPT, combines a protein phosphatase 1c (PP1c)-binding domain of MYPT1 with an optogenetic dimerizer, so that it allows light-dependent recruitment of endogenous PP1c to the plasma membrane. Blue-light illumination is sufficient to induce dephosphorylation of myosin regulatory light chains and a decrease in actomyosin contractile force in mammalian cells and *Xenopus* embryos. The OptoMYPT system is further employed to understand the mechanics of actomyosin-based cortical tension and contractile ring tension during cytokinesis. We find that the relaxation of cortical tension at both poles by OptoMYPT accelerated the furrow ingression rate, revealing that the cortical tension substantially antagonizes constriction of the cleavage furrow. Based on these results, the OptoMYPT system provides opportunities to understand cellular and tissue mechanics.

[1] National Institute for Basic Biology, National Institutes of Natural Sciences, 5-1 Higashiyama, Myodaiji-cho, Okazaki, Aichi 444-8787, Japan. [2] Exploratory Research Center on Life and Living Systems (ExCELLS), National Institutes of Natural Sciences, 5-1 Higashiyama, Myodaiji-cho, Okazaki, Aichi 444-8787, Japan. [3] Department of Basic Biology, School of Life Science, SOKENDAI (The Graduate University for Advanced Studies), 5-1 Higashiyama, Myodaiji-cho, Okazaki, Aichi 444-8787, Japan. [4] Graduate School of Arts and Sciences, University of Tokyo, Komaba 153-8902 Tokyo, Japan. [5] Research Center for Complex Systems Biology, Universal Biology Institute, University of Tokyo, Komaba 153-8902 Tokyo, Japan. [6] Japan Science and Technology Agency (JST), PRESTO, 4-1-8 HonchoKawaguchi Saitama 332-0012, Japan. [7] IRCC International Research Collaboration Center, National Institutes of Natural Sciences, 4-3-13 ToranomonMinato-ku Tokyo 105-0001, Japan. [8] These authors jointly supervised this work: Yohei Kondo, Kazuhiro Aoki. ✉email: y-kondo@nibb.ac.jp; k-aoki@nibb.ac.jp

Actomyosin contractility underlies force generation in a wide range of cellular and tissue morphogenesis in animals. Prominent examples include the tail retraction of directionally migrating fibroblasts and the constriction of a contractile ring during cytokinesis[1,2]. The actin-rich cell cortex, a thin network underneath the plasma membrane, is also relevant to the actomyosin contractility involved in maintaining cell morphology; namely, the actomyosin contractility at the cell cortex not only tunes mechanical rigidity, but also renders cells rapidly deformable as manifested in cell division and amoeboid migration[3,4]. Thus, it is of critical importance to disentangle the mode of action of actomyosin in order to understand how cells generate force and shape their morphology.

The actomyosin contractility in nonmuscle cells is mainly attributed to the force generated by nonmuscle myosin II (NMII)[5]. NMII contains two heavy chains, two essential light chains, and two regulatory light chains[5,6]. The myosin regulatory light chains (MLCs) are phosphorylated by myosin light chain kinase (MLCK) and Rho-kinase (ROCK), thereby inducing a conformational change in NMII and increasing its motor activity[5]. Small chemical compounds have been widely used to perturb the actomyosin contractility, such as blebbistatin (an inhibitor for NMII ATPase activity), Y-27632 (a ROCK inhibitor), and ML-7 (an MLCK inhibitor)[7–9]. While these compounds have allowed researchers to better understand the function of NMII, it is still technically challenging to control their actions at the subcellular resolution because of their rapid diffusion.

To overcome this limitation, recent efforts have been devoted to the development and application of optogenetic tools to manipulate cell signaling related to actomyosin contractility[10]. The most popular approach is to control the activity of RhoA, a member of Rho family small GTPases. Light-induced recruitment of RhoGEF triggers RhoA activation, which in turn activates ROCK and inactivates myosin light chain phosphatase (MLCP)[11–13] (Fig. 1a). These reactions eventually induce myosin light chain phosphorylation, followed by an increase in the actomyosin contractility. It has been reported that local accumulation of RhoGEF by light increases contractile force at the subcellular scale[11,13,14]. These technologies allow activation of NMII at the equator and the induction of partial constriction in rounded cells in metaphase[11]. Valon et al. further demonstrated that trapping of overexpressed RhoGEF to the outer membrane of mitochondria resulted in a decrease in actomyosin contractility[13]. In addition, depletion of PI(4,5)P$_2$ at the plasma membrane by optogenetic membrane translocation of 5-phosphatase OCRL has been shown to modulate cell contractility and inhibit apical constriction during *Drosophila* embryogenesis[15]. Although many of these tools enhance actomyosin contractility through RhoA or phospholipids, tools that reduce actomyosin contractility below the basal level have not yet been developed.

Here, we report an optogenetic tool to directly inactivate NMII; the system, called OptoMYPT, is designed to recruit an endogenous catalytic subunit of type Ic phosphatase (PP1c) to the plasma membrane with light, thereby dephosphorylating and inactivating NMII. We demonstrate that MLCs are dephosphorylated and the traction force exerted by cells is reduced at the local area where a blue light is illuminated. In *Xenopus* embryos, OptoMYPT decreases the tension along the cell–cell junction, leading to deformation of the cell–cell junction. Moreover, this system is applied to the mechanics of cytokinesis to understand how and to what extent actomyosin-based cortical tension antagonizes contractile ring tension and contributes to the cleavage furrow ingression rate.

## Results

### Design of an OptoMYPT system for reducing intracellular contractile force.

To manipulate the intracellular contractile force, we focused on MLCP, which is composed of three subunits, a catalytic subunit (PP1c), a regulatory subunit (MYPT1), and a smaller subunit of 20-kDa (M20)[16]. MYPT1 contains a PP1c-binding domain (PP1BD) and myosin heavy chain (MHC)-binding domain (Fig. 1b). MYPT1 holds PP1c through the PP1BD, and recruits it to NMII to dephosphorylate MLC, leading to the inactivation of NMII. Phosphorylated MLC is mainly localized near the plasma membrane such as at cortical actin and stress fibers, where the NMII exerts mechanical force[17].

Our strategy for the reduction of contractile force is based on inducing membrane translocation of the PP1BD in MYPT1 with light, resulting in the co-recruitment of endogenous PP1c at the plasma membrane and dephosphorylation of MLC. We refer to this system as the OptoMYPT system. It has been reported that the 1–38 amino acids (a.a.) in the PP1BD are particularly important for binding to PP1c, and that the 170–296 a.a. in the PP1BD serve as a phosphorylated MLC-binding domain[18]. As an optogenetic switch in this study, we mainly employed the improved Light-Induced Dimer (iLID) system, which binds to its binding partner, SspB, upon blue light illumination and dissociates from SspB under the dark condition[19]. The iLID-based OptoMYPT system consists of a light-switchable plasma membrane localizer, Stargazin-mEGFP-iLID, and an actuator, SspB-mScarlet-I-PP1BD, which is translocated to the plasma membrane for the co-recruitment of the endogenous PP1c with blue light (Fig. 1c). The Stargazin-mEGFP-iLID is suited for the subcellular protein recruitment, because the large N-terminal transmembrane anchor limits the diffusion of SspB proteins[20]. Alternatively, we developed a cryptochrome 2 (CRY2)-based OptoMYPT system, in which CRY2-mCherry-PP1BD was recruited to the plasma membrane with blue light through binding to the plasma membrane localizer CIBN-EGFP-KRasCT[21]. Since PP1c is known to dephosphorylate various substrates, it should be kept in mind that the OptoMYPT dephosphorylates non-MLC substrates such as ezrin-radixin-moesin (ERM) family of plasma membrane-actin cytoskeleton cross-linking proteins.

We first compared the efficacy of light-induced membrane translocation between three different lengths of PP1BDs: 1–38, 1–169, and 1–296 (hereafter referred to as MYPT38, MYPT169, and MYPT296, respectively). In line with the previous study[22], SspB-mScarlet-I-PP1BDs accumulated at the nucleus (Supplementary Fig. 1a, b). To circumvent this problem, the nuclear export signal (NES) was fused with the C terminus of the PP1BDs to export them to the cytoplasm (Supplementary Fig. 1a, b). We confirmed that Madin-Darby Canine Kidney (MDCK) cells exhibited the translocation of SspB-mScarlet-I from the cytoplasm to the plasma membrane upon blue light illumination as a control (Fig. 1d, e). SspB-mScarlet-I-MYPT169 showed the best membrane translocation in three differential lengths of PP1BDs (Fig. 1e and Supplementary Movie 1, left). We recognized that a small fraction of SspB-mScarlet-I-MYPT38 still resided in the nucleus (Fig. 1d, yellow arrowhead), and CRY2-mCherry-MYPT38 formed aggregates and puncta in a blue light-dependent manner for an unknown reason (Supplementary Fig. 2).

Next, we investigated whether PP1BDs of MYPT1 indeed bind to PP1c and recruit it to the plasma membrane. In control cells, PP1c fused with miRFP703 (PP1c-miRFP703), which was mainly localized at the nucleus, did not show any change in the subcellular localization upon blue light illumination (Fig. 1d, f). As expected, SspB-mScarlet-I-MYPT38, -MYPT169, and -MYPT296 demonstrated similar levels of translocation of

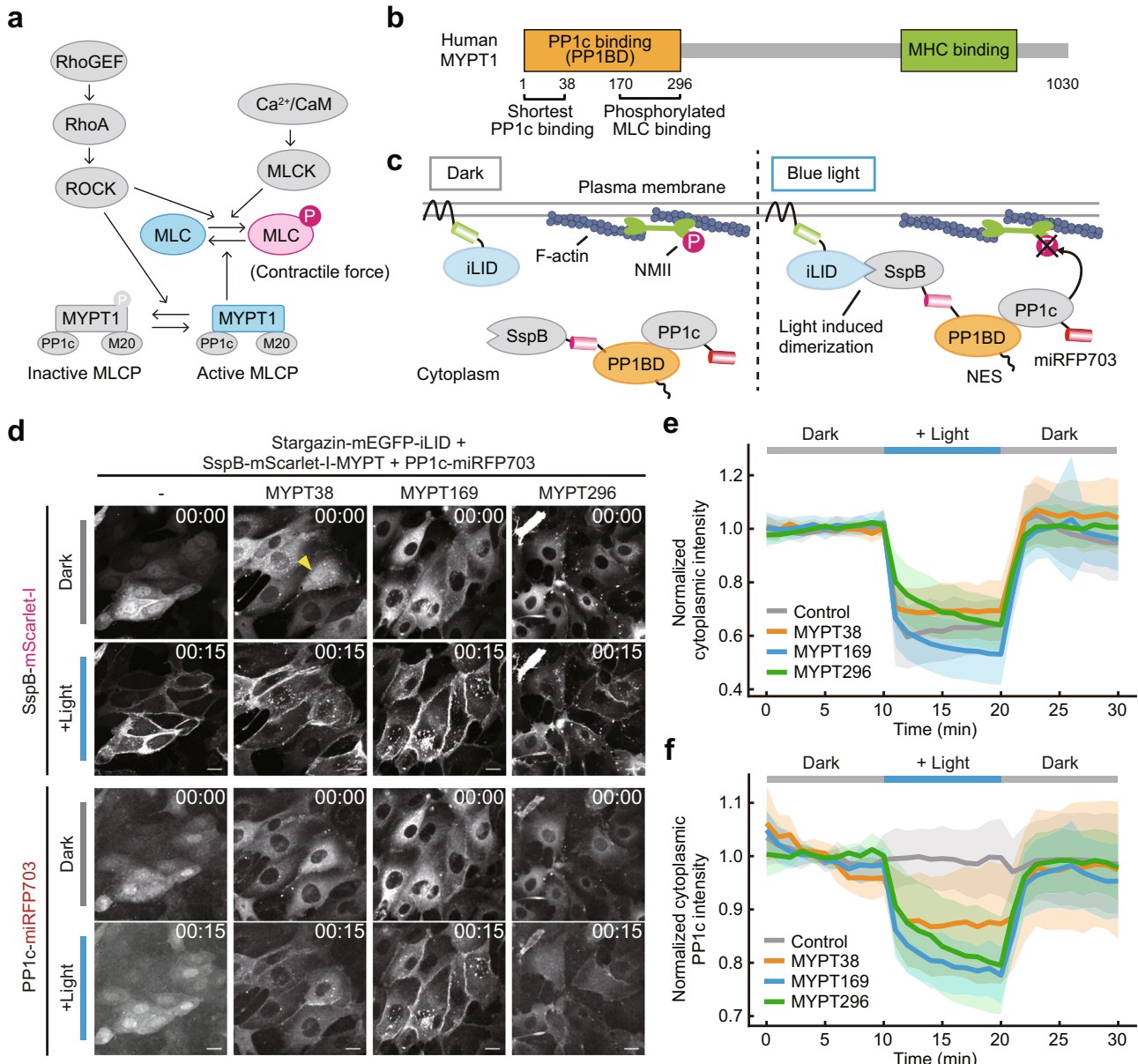

**Fig. 1 Development of the OptoMYPT system. a** Schematic illustration of the signaling pathway for the regulation of MLC phosphorylation. **b** Domain structure of human MYPT1. **c** Schematic illustration of the OptoMYPT system. Stargazin-mEGFP-iLID is anchored to the plasma membrane. Upon blue light illumination, the SspB-mScarlet-I-fused PP1c-binding domain (PP1BD) of MYPT1 translocates to the plasma membrane along with endogenous PP1c, and inactivates NMII at the cell cortex. **d** Representative images of the SspB-mScarlet-I or the indicated SspB-mScarlet-I-PP1BDs of MYPT1 (upper two rows) and simultaneously expressing PP1c-miRFP703 (lower two rows) in MDCK cells under the dark condition (the first and third rows) and blue light condition (the second and fourth rows). Stargazin-mEGFP-iLID was also expressed as a localizer in all experiments. The yellow arrowhead indicates a cell showing nuclear accumulation of SspB-mScarlet-I-MYPT38. Blue light was continuously illuminated from the top of the stage. Scale bar, 20 μm. **e** Quantification of the cytoplasmic fluorescence change in mScarlet-I in panel **d**. The average values (bold lines) are plotted as a function of time with the SD. $n = 15$ cells. **f** Quantification of the cytoplasmic fluorescence change in PP1c-miRFP703 in the indicated MDCK cells. The average values (bold lines) are plotted as a function of time with the SD. $n = 15$ cells.

PP1c-miRFP703 from the cytoplasm to the plasma membrane upon blue light illumination (Fig. 1d, f and Supplementary Movie 1, right). We also confirmed the plasma membrane translocation of the endogenous PP1c by immunofluorescence analysis (Supplementary Fig. 1c). During these experiments, we noticed that the cells expressing MYPT296 showed elongated membrane protrusions even under the dark condition (Supplementary Fig. 1a), which was similar to the morphology of cells treated with ROCK inhibitor or myosin inhibitor[23,24], while the cell size was not substantially different between cells expressing

PP1BDs of MYPT (Supplementary Fig. 1d). We evaluated the effect of the expression of PP1BDs on basal phosphorylation of MLC by western blotting analysis. The expression levels of SspB-mScarlet-I-MYPT169 were estimated relative to that of the endogenous MYPT, showing an ~12-fold increase in stable expression and 50-fold increase in transient expression under our experimental conditions (Supplementary Fig. 3). As expected from the cell morphology, MLCs were dephosphorylated in MDCK cells stably expressing MYPT296 (Supplementary Fig. 1e). This result is consistent with the previous report pointing out the

existence of the phosphorylated MLC-binding domain at the 170–296 a.a.[18], which may facilitate recruitment of PP1BD to NMII and dephosphorylation of MLC without blue light stimulation. Under these conditions, the ERM family proteins maintained their phosphorylation levels detected with pan-phospho-ERM antibody (Supplementary Fig. 1e), suggesting that the expression of PP1BD of MYPT has no impact on phosphorylation of the ERM proteins. Taken together, these results led us to conclude that MYPT169 is well suited for the OptoMYPT system.

**Light-induced dephosphorylation of MLC and membrane protrusion by OptoMYPT.** To evaluate whether the OptoMYPT dephosphorylates MLC in a blue light-dependent manner, we directly measured phosphorylated MLC with immuno-fluorescence. Given the design of OptoMYPT, we expected that the OptoMYPT dephosphorylates MLCs that are located beneath the plasma membrane upon light stimulation. Therefore, we focused on lamellipodia, in which a sheet-like thin structure with an actin meshwork propels the cell membrane in a myosin IIA-dependent manner[25–27]. The blue light was locally illuminated at the lamellipodia in MDCK cells for 30 min, followed by fixation and immunofluorescence staining with the anti-phospho-MLC antibody (Fig. 2a). We herein adopted a CRY2-based OptoMYPT system[21], because the slower dissociation kinetics of the CRY2-CIB system compared to that of the iLID-SspB system was preferable for this experiment. Local illumination of blue light induced spatially restricted recruitment of CRY2-mCherry and CRY2-mCherry-MYPT169 (Fig. 2b, left column). In addition, the local recruitment of CRY2-mCherry-MYPT169, but not CRY2-mCherry, attenuated the MLC phosphorylation (Fig. 2b, right column). The quantification of phosphorylated MLC fluorescence intensity in dark and light illuminated areas (Fig. 2a) revealed a partial but significant reduction in the phosphorylated MLC level (Fig. 2c). We further evaluated the dephosphorylation of MLC by biochemical studies. To this end, we established a doxycycline-inducible expression system of OptoMYPT proteins, because when PP1BDs of MYPT were stably expressed over the long-term they tended to form aggregates. We found that the global blue light illumination partially decreased the phosphorylated MLC level, based on western blotting analysis in MDCK cells expressing OptoMYPT proteins (Supplementary Fig. 4). This partial reduction of MLC phosphorylation by OptoMYPT was probably due to the limited accessibility of PP1BD of MYPT to MLCs; OptoMYPT could reach MLCs that were located beneath the plasma membrane, such as lamellipodia, but not at the sites far from the plasma membrane. Note that just OptoMYPT expression induced a slight decrease in phosphorylated MLC levels even before blue light irradiation (Supplementary Fig. 4), implying that PP1c is partially activated by overexpression of MYPT169 with the doxycycline-inducible expression system.

The depletion of myosin IIA, blebbistatin treatment, or overexpression of the non-phosphorylatable form of MLC has been shown to result in membrane protrusion[25–28]. These reports prompted us to examine whether dephosphorylation of MLC by OptoMYPT would induce membrane protrusion. The control MDCK cells that expressed SspB-mScarlet-I, Stargazin-mEGFP-iLID, and Lifeact-miRFP703 demonstrated local accumulation of SspB-mScarlet-I by blue light illumination, but did not show the morphological change (Fig. 2d and Supplementary Movie 2, upper). On the other hand, the OptoMYPT-expressing MDCK cells reproducibly showed peripheral membrane protrusions in the blue-light exposed area (Fig. 2e and Supplementary Movie 2, lower). We evaluated light-induced membrane protrusion with a kymograph and time-course graph, which showed the movement

of the cell edge upon blue light illumination in OptoMYPT-expressing cells (Fig. 2f, g). The protruding membrane was subsequently maintained under dark conditions (Fig. 2f, g). Interestingly, we often recognized membrane retraction on the opposite side of the blue-light illumination area (Fig. 2h, arrowhead, Supplementary Movie 3), consistent with regulation of cell polarity by membrane tension[29]. Furthermore, the global, blue-light illumination of MDCK cells expressing OptoMYPT induced membrane protrusions mainly from lamellipodial regions (Supplementary Fig. 5a and Supplementary Movie 4). Similarly, in NIH-3T3 cells, the local illumination of blue light induced membrane protrusions from the pre-existing lamellipodia (Supplementary Fig. 5b, c, inset 1 and Supplementary Movie 5), but did not induce the change in stress fiber formation and membrane protrusions from the site in close proximity to stress fibers (Supplementary Fig. 5b, c, inset 2). This result suggests that MLCs on stress fibers probably escape the dephosphorylation by OptoMYPT because they are far from the plasma membrane. Taken together, we concluded that the OptoMYPT system dephosphorylates MLCs located beneath the plasma membrane by light stimulation and induces membrane protrusions.

**OptoMYPT reduced the traction force in migrating cells and tension at the cell–cell junction in *Xenopus* embryos.** Next, we examined whether the decrease in MLC phosphorylation by OptoMYPT affects actomyosin-based contractile force. To do this, traction force microscopy was applied to randomly migrating MDCK cells expressing the OptoMYPT system. The cells were seeded on polyacrylamide gel containing infra-red fluorescence beads, so that we could infer the traction force by the displacement of fluorescence beads and the mechanical properties of the polyacrylamide gel (Fig. 3a). The blue light was locally focused on the lamellipodial region, where the cells were generating strong traction force. SspB-mScarlet-I (control) and SspB-mScarlet-I-MYPT169 (OptoMYPT) were successfully recruited to the locally illuminated area (Fig. 3b, c, upper panels). In contrast to the control cells, the cells expressing the OptoMYPT system showed a decrease in traction force after blue light illumination (Fig. 3d, lower panels, Supplementary Movie 6). These results indicate that the OptoMYPT system can dephosphorylate MLCs by local blue light illumination, leading to a reduction of traction force in randomly migrating cells.

We further applied OptoMYPT to the in vivo system by using *Xenopus laevis* embryos. For this purpose, we observed the animal pole of stage 12 gastrula embryos, which generate relatively strong tension at the cell–cell junctions[30]. The embryos expressing SspB-mScarlet-I or SspB-mScarlet-I-MYPT169 together with Stargazin-mEGFP-iLID and Lifeact-miRFP703 demonstrated rapid plasma membrane translocation of SspB-mScarlet-I or SspB-mScarlet-I-MYPT169 in response to blue light illumination, respectively (Fig. 3e, upper panels and Supplementary Movies 7, left). In embryos expressing OptoMYPT, the cell–cell junctions became wavy in shape by blue light illumination, suggesting decreased actomyosin contractility (Fig. 3f, lower panels and Supplementary Movies 7, right)[31–33]. We quantified the temporal changes in waviness at the cell–cell junctions and found that waviness increased significantly in the cells expressing OptoMYPT constructs after blue light illumination (Fig. 3g and Supplementary Fig. 6a–c).

To directly validate the decrease in actomyosin contractility at the cell–cell junctions, we combined laser ablation with optogenetic experiments. This is because the tension along the cell–cell junction can be estimated by measuring the recoil velocity of the cell–cell junction after laser ablation[34,35]. For the

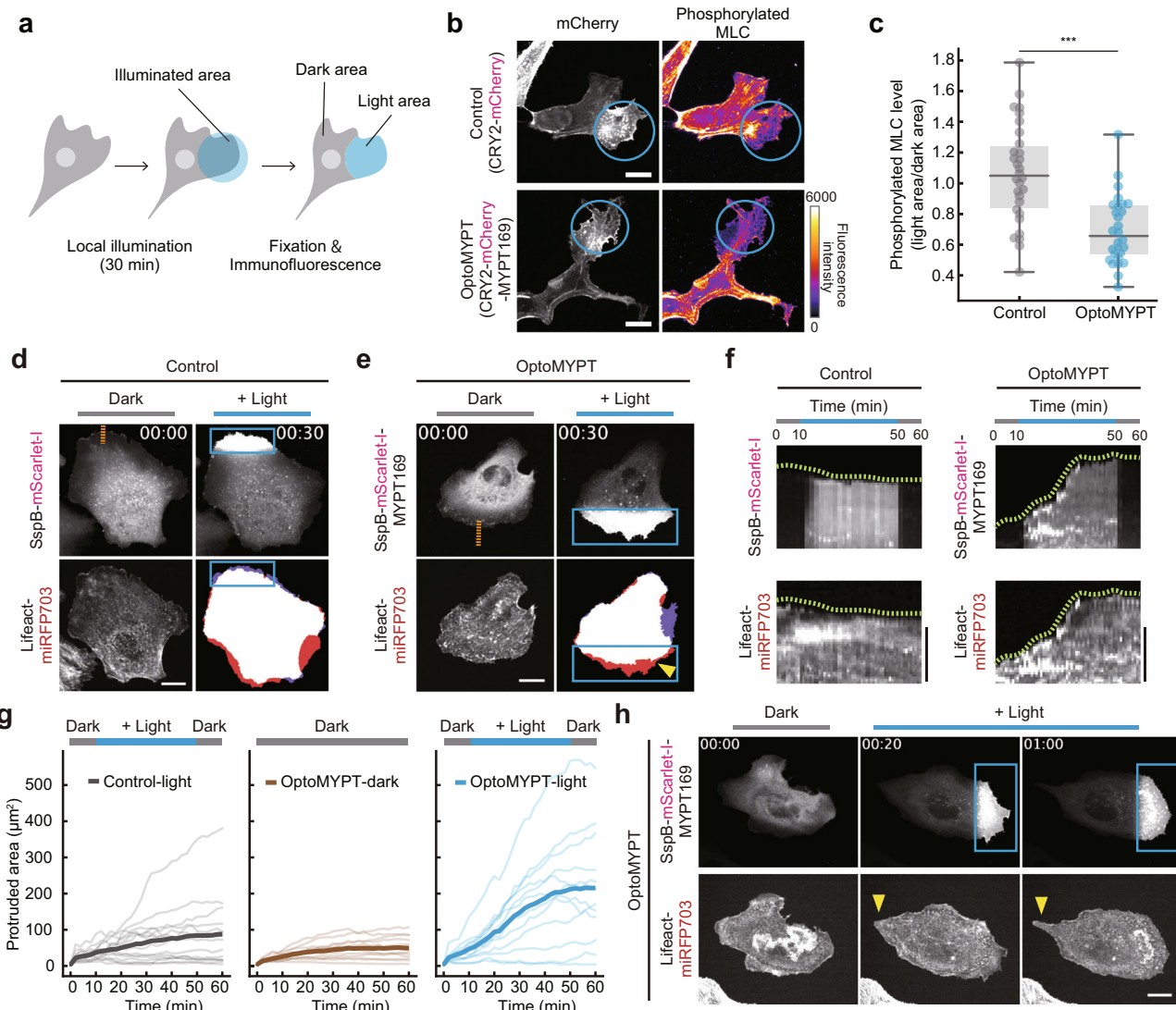

**Fig. 2 Light-induced dephosphorylation of MLC and membrane protrusion by OptoMYPT. a** Schematic illustration of an experimental procedure to quantify phosphorylated MLC levels. The blue light was locally focused on the lamellipodial area in each cell, followed by fixation and immunostaining. **b** Immunofluorescence analysis of MLC phosphorylation after blue-light illumination. The upper and lower images show MDCK cells expressing CRY2-mCherry and CRY2-mCherry-MYPT169, respectively. CIBN-EGFP-KRasCT was also expressed as a localizer in both experiments. The blue circular region was illuminated with 500 msec of blue light at 2 min intervals for 30 min. Scale bar, 10 μm. **c** The phosphorylated MLC level was quantified by dividing the mean fluorescence intensity of the light area by that of the dark area in panel **a**, and shown as a box plot. $n = 30$ and 28 cells for the control and OptoMYPT, respectively. ***$p = 8.4 \times 10^{-8}$ (two-sided Brunner–Munzel test)[74]. **d**, **e** Simultaneous visualization of SspB-mScarlet-I (**d**) or SspB-mScarlet-I-MYPT169 (**e**) with F-actin (Lifeact-miRFP703) in MDCK cells. The blue rectangular regions were illuminated with 500 msec blue light every 20 sec. The right bottom image shows a binary image reconstructed from Lifeact-miRFP703 images; red and purple areas represent protruding and retracting areas, respectively. The yellow arrowheads depict membrane protrusion. Scale bar, 20 μm. **f** Kymographs were drawn along the orange dashed lines in panels **a** and **b**. Green dashed lines show cell boundaries. Scale bar, 5 μm. **g** Quantification of the local protruding areas under the indicated conditions. The total protruded area was calculated by subtracting the cell area in the illuminated region at $t = 0$ from that at each time point. Local blue light was illuminated from 10 to 50 min. The thin and bold lines indicate the individual and averaged data, respectively. $n = 15$, 13, and 12 cells for Control-light (blue light illumination), OptoMYPT-dark (dark condition), OptoMYPT-light (blue light illumination), respectively. **h** Representative images of the induction of membrane retraction (yellow arrowhead) on the opposite side of the blue-light illuminated area. The blue rectangular regions were locally illuminated with 500 msec blue light every 20 s. Scale bar, 20 μm.

convenience, the ablation point was determined based on the membrane-targeted mEGFP signal at the cell–cell junctions. To prevent membrane translocation of SspB-mScarlet-I-MYPT169 by excitation light for mEGFP, mEGFP-KRasCT, which is a plasma membrane-localized protein incapable of recruiting SspB proteins, was co-expressed instead of Stargazin-mEGFP-iLID and was used as an "OptoMYPT-no translocation" control (Fig. 3h). In these experiments, the cells were 20 min pre-illuminated and then illuminated by the mEGFP excitation blue

light for laser ablation experiments. To avoid potential contributions of the initial junction lengths to recoil, cell–cell junctions of similar lengths were selected for the laser ablation experiments (Supplementary Fig. 6d). We tested the following three conditions; Control (SspB-mScarlet-I and Stargazin-mEGFP-iLID), OptoMYPT-no translocation (SspB-mScarlet-I-MYPT169 and mEGFP-KRasCT), and OptoMYPT (SspB-mScarlet-I-MYPT169 and Stargazin-mEGFP-iLID). The center of the cell–cell junctions was ablated, and the displacement was

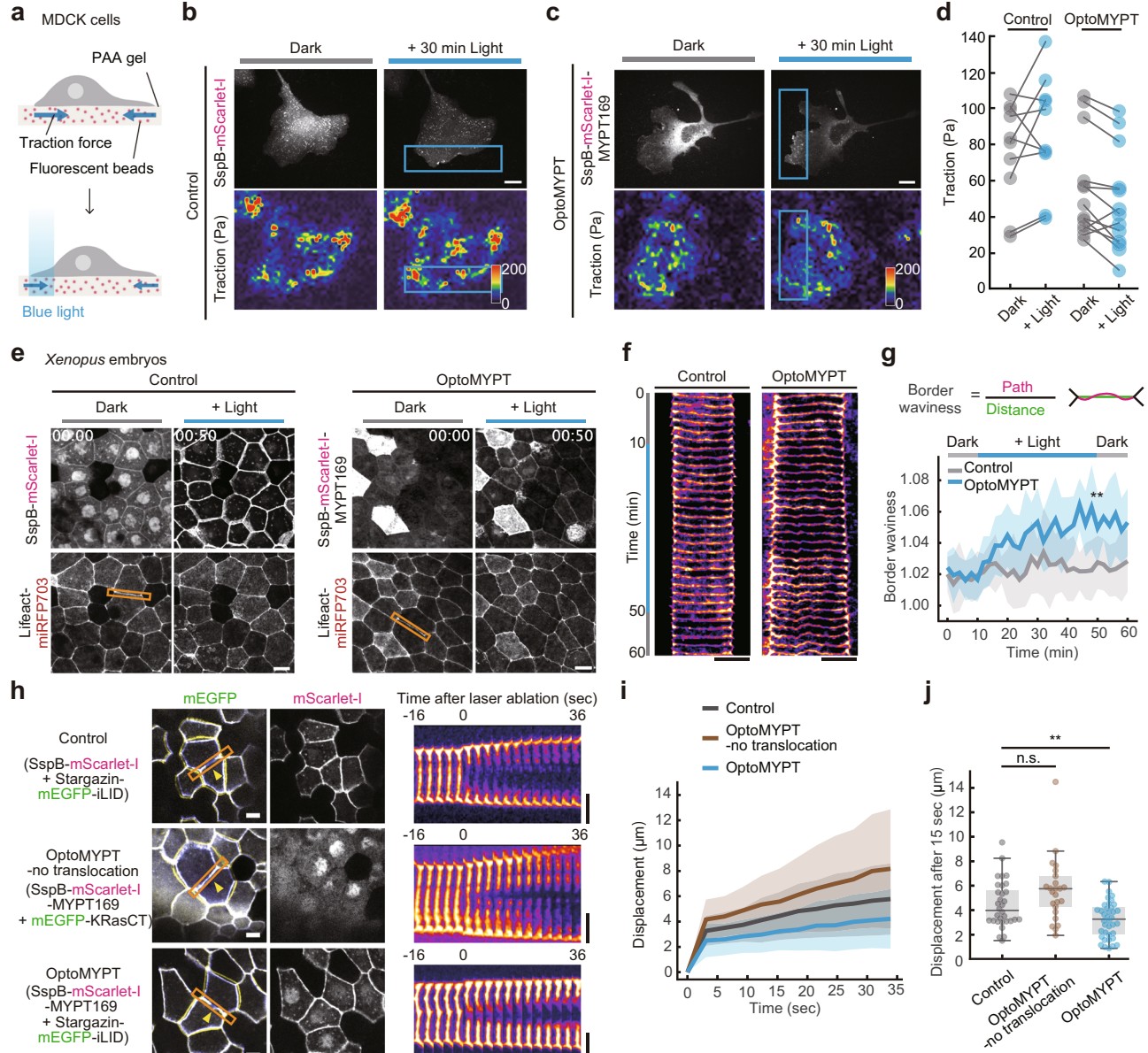

**Fig. 3 OptoMYPT reduced the traction force in migrating cells and tension at the cell–cell junction in *Xenopus* embryos. a** Schematic of the traction force microscopy. **b**, **c** Traction force measurement in an MDCK cell expressing SspB-mScarlet-I (**b**), and SspB-mScarlet-I-MYPT169 (**c**) with Stargazin-mEGFP-iLID. The blue rectangular regions were locally illuminated with 500 msec of blue-light at 20 s interval for 30 min. Traction force (Pa) is represented as a pseudocolor. **d** Quantification of the traction force. $n = 10$ and 13 cells for Control and OptoMYPT cells, respectively. **e** Simultaneous visualization of SspB-mScarlet-I (left) or SspB-mScarlet-I-MYPT169 (right) with F-actin (Lifeact-miRFP703) in *Xenopus* embryos at gastrula stages. **f** Kymographs were drawn along the orange boxed areas in panels **e**. **g** Quantification of the border waviness calculated by dividing the actual length of the cell-cell junction by the distance. Blue light was illuminated for a total of 10–50 min. $n = 15$ cells from three embryos for both the Control and OptoMYPT experiment. \*\*$p = 0.006$ by two-tailed Student's *t*-test at 50 min. **h** Laser ablation of the animal pole at gastrula stages. Representative images of plasma membranes visualized by Stargazin-mEGFP-iLID (upper and lower) or mEGFP-KRasCT (middle) in the left column and SspB-mScarlet-I (upper) and SspB-mScarlet-I-MYPT169 (middle and lower) in the right column. The images at $t = 16$ s (blue) before and 36 sec (yellow) after ablation were overlaid with the image at $t = 0$ s (gray). The yellow arrowheads depict the ablation points. Blue light was continuously illuminated from the top of the embryo 20 min before imaging and continued to be illuminated through the objective lens every 3 s to visualize cell-cell junctions. The right kymographs were drawn along the orange boxed areas in panel **h**. **i** Time-course of the displacement after ablation. **j** Displacement 15 s after ablation. \*\*$p = 0.004$ by two-tailed Student's *t* test. **g**, **i** The average values (bold lines) are plotted as a function of time with the SD. **i**, **j** $n = 32$, 24, 38 cells from six embryos examined over two independent batches for Control, OptoMYPT-no translocation and OptoMYPT, respectively. All scale bars, 20 μm.

measured from the subsequent change in the distance between cell–cell junctions (Fig. 3h and Supplementary Movie 8). The recoil velocity of the cell-cell boundary was significantly slower in OptoMYPT cells (0.81 ± 0.42 μm/min) than in Control cells (1.06 ± 0.38 μm/min) and OptoMYPT-no translocation cells

(1.35 ± 0.50 μm/min) (Fig. 3i, j and Supplementary Fig. 6e), indicating the reduced tension at the cell–cell junction by OptoMYPT. These results demonstrate that the OptoMYPT system is applicable in vivo and decreases actomyosin contractility in a multicellular context.

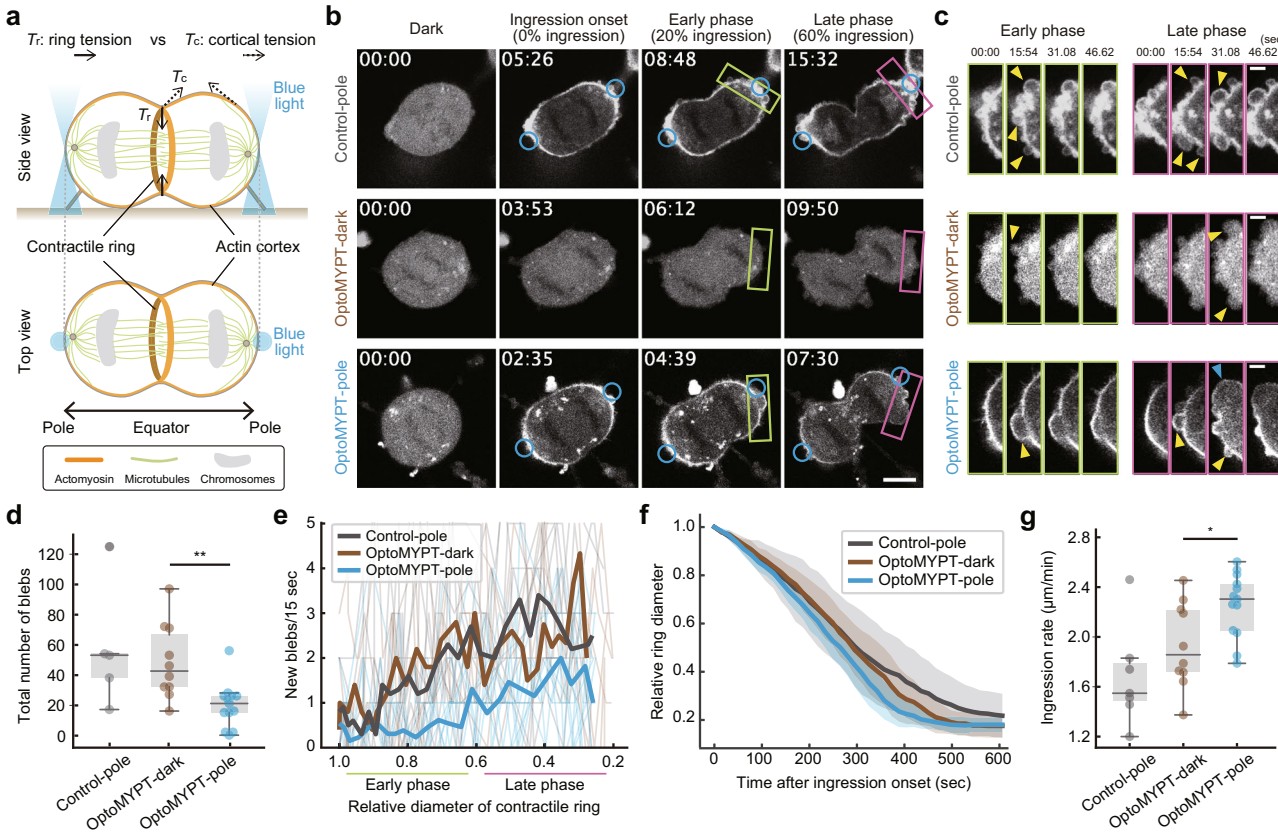

**Fig. 4 Examination of the actomyosin-based cortical tension during cytokinesis with OptoMYPT. a** Schematic illustration of cytokinesis in animal cells. Solid and dashed arrows indicate ring tension and cortical tension, respectively. Orange, green, and gray objects indicate actomyosin, microtubules, and chromosomes, respectively. Blue light is focused on the poles on both sides. **b** Representative images of SspB-mScarlet-I (upper) or SspB-mScarlet-I-MYPT169 (middle and lower) in MDCK cells during cytokinesis. The blue circular regions were locally illuminated with blue light every 3.11 s. Middle panels represent the cytokinesis of a cell under dark conditions. Scale bar, 10 μm. **c** Inset images of polar blebbing (the green and magenta boxed regions in panel **b**, representing the early and late phases, respectively). Yellow and blue arrowheads indicate small and large new blebs per stack, respectively. Scale bar, 3 μm. **d**, **e** Quantification of the total number of blebs during cytokinesis, shown as a box plot (**d**), and of the number of new blebs emerged within 15.54 s, shown as a line graph (**e**), in which thin and bold lines indicate individual and averaged data, respectively. $n = 5$, 10, and 13 cells for Control-pole, OptoMYPT-dark, and OptoMYPT-pole, respectively. $**p = 0.002$ (two-tailed Student's $t$ test). **f**, **g** Quantification of the furrow ingression rate after ingression onset. Averaged relative diameters are plotted as a function of time with the SD (**f**). The ingression rate was estimated by calculating the slope of the ingression rate from 1.0 to 0.6 in panel **f**, and shown as a box plot (**g**). $n = 7$, 10, and 13 cells for Control-pole, OptoMYPT-dark, and OptoMYPT-pole, respectively. $*p = 0.017$ (two-tailed Student's $t$ test).

**Acceleration of the ingression rate of cleavage furrows during cytokinesis by the optical relaxation of cortical tension.** We next applied the OptoMYPT system to elucidate the mechanical regulation of the actin cortex during cytokinesis. In this process, actin, NMII, and cross-linkers constitute a contractile ring in the equatorial plane, and generate force to divide a cell into two daughter cells by constriction[1,36] (Fig. 4a, solid arrows). On the other hand, the tension developed in cortical actomyosin counteracts the force in the contractile ring[37–42] (Fig. 4a, dashed arrows). Thus, to advance the constriction, the contractile ring has to overcome the resistance of the actin cortex. There are two mechanisms for driving the cytokinesis; increasing tension in the ring and mechanical weakening of the cell cortex. The latter mechanism, so-called "polar relaxation", through, for example, NMII removal from the polar region, is required for proper cytokinesis[37–42], but genetic and pharmacological inhibition of cortical actomyosin often induce cytokinetic failure[43–46]. To elucidate the underlying mechanics, the strength of cortical tension has been directly measured using atomic force microscopy, microaspiration, and laser ablation methods[43,47,48]. However, clarifying the contribution of cortical tension is still technically challenging due to the highly dynamic nature of cytokinesis.

To quantitatively address the cytokinetic mechanics, we expected that optogenetic perturbation of the cortical tension by the OptoMYPT system would be a potentially effective approach. Blue light was locally and repeatedly illuminated at both poles from the onset of chromosome segregation (Fig. 4a). SspB-mScarlet-I and SspB-mScarlet-I-MYPT169 were trapped in the polar region (Control-pole and OptoMYPT-pole, respectively) (Fig. 4b and Supplementary Movies 9, left and right). Because the overexpression of OptoMYPT might reduce basal actomyosin activity (Supplementary Fig. 4), cells expressing SspB-mScarlet-I-MYPT169 under a dark condition throughout cytokinesis (OptoMYPT-dark) were also used as another control (Fig. 4b and Supplementary Movie 9, middle).

We focused on the dynamics of membrane blebbing at the polar cortex during cytokinesis, because bleb formation requires the intracellular pressure generated by cortical tension to be high enough to overcome membrane-cortex anchoring and surface tension of the plasma membrane[48–50]. In our experiments, the level of membrane bleb formation following the onset of cleavage furrow ingression was lower in OptoMYPT-pole cells than Control-pole or OptoMYPT-dark cells (Fig. 4c, d). In addition, we found that the local activation of OptoMYPT altered the onset

and size of membrane bleb formation during cytokinesis (Fig. 4e and Supplementary Fig. 7). First, the onset of blebbing was delayed in OptoMYPT-pole cells, while Control-pole and OptoMYPT-dark cells showed blebbing from the early phase of ring constriction. Second, in the later phase of cytokinesis, OptoMYPT-pole cells exhibited large blebs (Supplementary Fig. 7), although the bleb counts were still smaller than the control cases. We should note that the OptoMYPT activation might weaken the membrane-cortex linkage, such as by ERM deactivation. However, this weakening of the membrane-cortex linkage has been shown to render the OptoMYPT-pole cells more prone to bleb formation[40,50]. Therefore, the absence of blebbing in the early phase indicates reduced tension in OptoMYPT-pole cells independent of whether membrane-cortex linkage is weakened or not. On the other hand, the larger blebs of the OptoMYPT-pole cells in the late phase, possibly initiated by excess intracellular pressure coming from ring constriction, might be explained by both the weakening of the membrane-cortex linkage and inefficient bleb retraction (see the Discussion section for more details).

Next, we investigated the effects of OptoMYPT on the dynamics of ring constriction. We measured the furrow ingression rate under each condition to gain insight into how the cortices affected the furrow ingression. The ingression rate of the cleavage furrow was significantly higher in OptoMYPT-pole cells ($2.26 \pm 0.25$ μm/min) than in Control-pole cells ($1.55 \pm 0.20$ μm/min) and OptoMYPT-dark cells ($1.93 \pm 0.33$ μm/min) (Fig. 4f, g). These results highlight the negative contribution of the cortices to the cleavage furrow ingression. Of note, the ingression rate of OptoMYPT-dark cells was slightly higher than that of Control-pole cells (Fig. 4g), suggesting the basal effects of OptoMYPT on MLC phosphorylation even under the dark condition. To quantify how the OptoMYPT activation modified the cortical tension, we adopted a coarse-grained physical model describing the mechanics of cytokinesis[37,38,49]. The cortical tension represents both the myosin-generated tension and the viscoelastic response of the cortical cytoskeleton. We note that the OptoMYPT activation might have modulated not only the myosin-generated tension but also the viscoelastic response through, for example, inhibition of the cross-linking activity of NMII. In this model, the furrow ingression rate ($v$) is considered to be proportional to the difference between the contractile force of the ring ($F_r$) and the resisting force exerted by the cortices ($F_c$), $v \propto F_r - F_c$ (see the Materials and methods section, and Supplementary Note 1) (Supplementary Fig. 8). Based on the experimental data with the coarse-grained physical model, we estimated that the resisting force exerted by the cortices corresponds to 15%~31% of the ring tension during cytokinesis (see the Supplementary Note 1).

Finally, we varied the local activation patterns of OptoMYPT in dividing cells. Sedzinski et al. have demonstrated that the perturbation of the cortex at one pole of dividing cells by laser ablation or local application of an actin-depolymerizing drug leads to cytokinetic shape oscillations[49]. To reproduce this report, we illuminated only one of the polar regions by blue light throughout cytokinesis (OptoMYPT-single pole). As expected, we often observed extensive cell shape oscillation with the back-and-forth motion of separated chromosomes (Fig. 5a). In the early event of cell shape oscillation, a large bleb was formed from the pole irradiated with blue light (Fig. 5a, yellow arrowhead). This result is consistent with a report showing that cytokinetic oscillation is triggered by the retraction of large blebs[43]. We classified oscillation into three types: normal cytokinesis (no oscillation occurs), small oscillation (the daughter chromosomes oscillate slightly) (e.g. Supplementary Movie 9, Control-pole), and large oscillation (both chromosomes enter one daughter cell) (Fig. 5a and Supplementary Movie 10). OptoMYPT-single pole experiments increased the fraction of cells showing "large

oscillation" (Fig. 5b). OptoMYPT-dark seems to have a lower percentage of oscillation than Control-pole (Fig. 5b), possibly due to the decrease in basal cortical tension by the expression of OptoMYPT. We also conducted an experiment in which the equatorial plane of dividing cells was irradiated with blue light, because it has been reported that the actomyosin-based contractility is associated with the constriction of the contractile ring[1,36]. However, blue light illumination at the equatorial plane of dividing cells had no influence on the ingression rate of cleavage furrow (see the Discussion section for more details) (Supplementary Fig. 9).

## Discussion

In this study, we developed an optogenetic tool, OptoMYPT, and demonstrated light-dependent relaxation of cellular forces at the subcellular level. The OptoMYPT is substantially different from existing optogenetic tools related to cell mechanics in two ways. First, the OptoMYPT reduces contractile forces below the basal level, and therefore provides additional flexibility for in situ control of actomyosin contractility and cellular morphology. Second, the OptoMYPT directly regulates NMII through MLC dephosphorylation, whereas optogenetic modulation of RhoA activity or $PI(4,5)P_2$ may affect pathways other than NMII, since RhoA and $PI(4,5)P_2$ are known to control various downstream effectors such as ROCK, mDia, and the ERM proteins[51–53].

Using the OptoMYPT system, we experimentally revealed the negative contribution of the cortical tension to the cleavage furrow ingression rate during cytokinesis; i.e., the decrease in cortical tension at both poles by OptoMYPT accelerates furrow ingression (Figs. 4 and 5c). It has been reported that the reduction of cortical tension by laser ablation in the polar region decelerates cleavage furrow ingression in cytokinesis of *C. elegans* embryos[54]. This discrepancy could be due to the difference in the force balance between the pole and equator; in *C. elegans* embryos, NMII is actively removed from the polar region and accumulates at the equator due to the cortical flow, and thus cortical tension is much weaker than ring tension[42,55]. Meanwhile, our results indicate that cortical tension in cultured mammalian cells is comparable to contractile ring tension (Fig. 4), which is in good agreement with the previous work[49]. Such a high cortical tension is advantageous because it confers shape stability to mitotic cells[49]. This has been corroborated in a recent paper demonstrating that high cortical stiffness in cancer cells allows them to divide in a confined environment[56]. The benefit of cortical tension is not required in *C. elegans* embryos, because they are covered and protected by a rigid eggshell. However, high cortical tension is a double-edged sword, because excessive cortical tension induces cytokinetic shape oscillation and abscission failure[43,57,58]. The estimated cortical tension relative to ring tension ($F_c/F_r$), ~20%, may achieve a balance between morphological maintenance and timely cytokinesis.

We also focused on the dynamics of membrane blebbing during cytokinesis (Figs. 4 and 5). Membrane blebbing is initiated by local rupture of the actin cortex and/or detachment of the actin cortex from the plasma membrane[50]. The growth of membrane blebs requires a condition under which hydraulic pressure caused by actomyosin-based cortical tension overcomes plasma membrane tension[48]. Using the OptoMYPT system, we showed the absence of membrane blebbing in OptoMYPT-pole cells in the early phase of furrow ingression (Fig. 4e). On the other hand, Rodrigues et al.[40] reported that optogenetic inactivation of moesin, a member of the ERM family proteins, induces a bleb from the actin cortex during metaphase. These opposite effects support that, in OptoMYPT-pole cells, the reduction in cortical tension prevails over the weakening of membrane-cortex

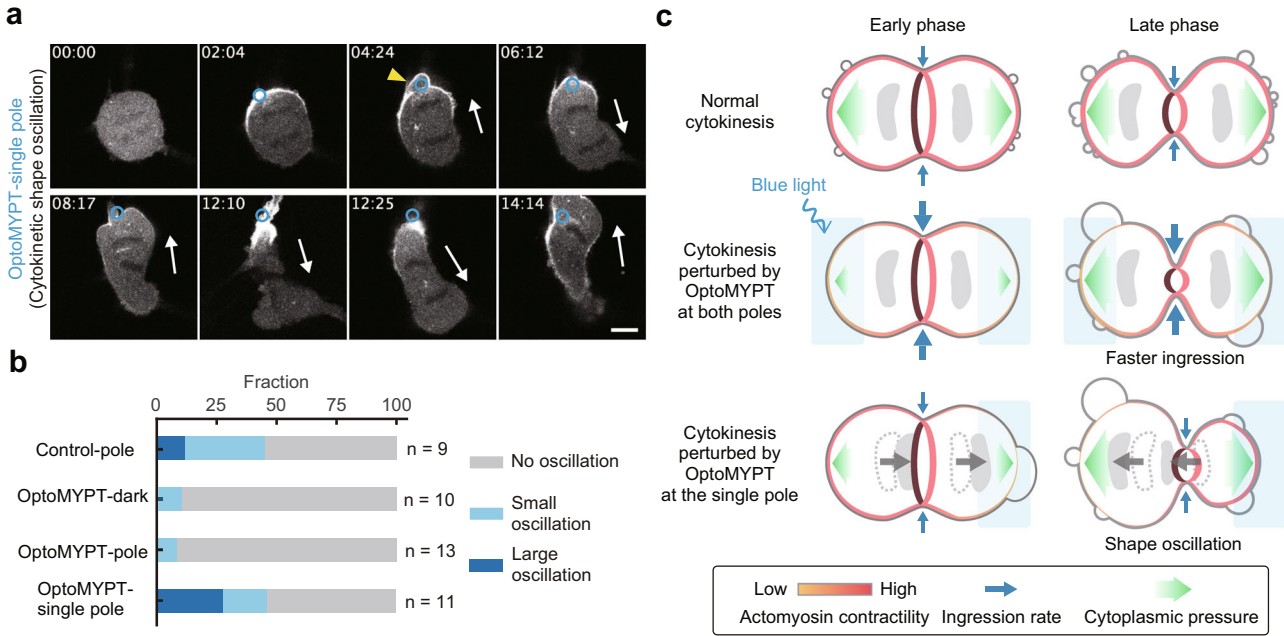

**Fig. 5 OptoMYPT induced shape instability during cytokinesis. a** Representative images of SspB-mScarlet-I-MYPT169 during cytokinesis upon illumination to the single pole. The blue circular regions were locally illuminated with blue light every 3.11 s. The yellow arrowhead and white arrow indicate a large bleb just before spindle oscillation and the direction of chromosome motion, respectively. Scale bar, 10 μm. **b** Quantification of the fraction of the cytokinetic dynamics under the indicated conditions. The small oscillation and large oscillation are defined as a slight oscillation of the spindle and an oscillation of two daughter chromosomes entering one of the cells, respectively. **c** Schematic model of mechanical regulation of the contractile ring and the actin cortex during cytokinesis. In control cells, high cortical tension acts as a decelerator of the ring constriction. The high cortical tension and cytoplasmic pressure induce blebbing from the early phase of cytokinesis. The increased cytoplasmic pressure associated with ring constriction in the late phase is released by the increased number of blebs. In OptoMYPT-expressing cells illuminated by blue light at both poles, bleb formation is suppressed in the early phase of cytokinesis and the cleavage furrow ingression is accelerated due to the decrease in the cortical tension. As cleavage furrow ingression progresses, large blebs emerge due to the increased cytoplasmic pressure in the late phase of cytokinesis. In OptoMYPT-expressing cells illuminated by blue light at a single pole, the reduced cortical tension at one side leads to the formation of a large bleb and the cytoplasmic flow. The retraction of the bleb induces cell shape oscillation accompanied with the back-and-forth movement of chromosomes.

linkage which is a possible side effect of OptoMYPT. However, even in OptoMYPT-pole cells, we reproducibly observed membrane blebbing in the late phase (Fig. 4e). Recently, Wang et al.[59] have shown that the increase in the extracellular osmotic pressure, i.e., relative decrease in the cytoplasmic pressure, induces the delayed onset of blebbing during cytokinesis. These results support the idea that the ring constriction gradually increases cytoplasmic pressure, which causes bleb formation in the later phase.

There still remain some issues to be addressed with respect to the OptoMYPT. First is the issue of substrate specificity in OptoMYPT. We could not exclude the possibility that the OptoMYPT dephosphorylates additional substrates other than MLC. In addition, the expression of PP1BD of MYPT affected the localization of endogenous PP1c (Fig. 1d and Supplementary Fig. 1c). Some proteins are dephosphorylated by PP1c in the nucleus[60], and therefore the nuclear exclusion of PP1c by OptoMYPT may disturb its nuclear function. However, based on the fact that the expression of MYPT296 did not dephosphorylate ERM proteins (Supplementary Fig. 1e), it seems somewhat unlikely that OptoMYPT dephosphorylates and inactivates the ERM proteins. The next issue is the localizer of optogenetic switches. In the current OptoMYPT system, dephosphorylation of MLC was induced by recruiting MYPT169 to the plasma membrane upon illumination with blue light. It is plausible that the OptoMYPT dephosphorylates and inactivates only the NMII existing in the vicinity of the plasma membrane, such as in the lamellipodial region (Fig. 2) and at cell–cell junctions in *Xenopus* embryos (Fig. 3). Meanwhile, our data suggest that OptoMYPT is incapable of dephosphorylating MLCs that are located away from the plasma membrane and/or that are incorporated in the highly bundled actin fibers, such as stress fibers (Supplementary Fig. 5b, c) and the contractile ring (Supplementary Fig. 9). Especially in the latter case, it is technically difficult to confine local activation of OptoMYPT at the contractile ring by conventional confocal microscopy, because of the diffusion of OptoMYPT proteins and the shape of the point spread function on the defocused planes with a high numerical aperture objective lens (Supplementary Fig. 9). These effects might lead to the dephosphorylation of MLC not only in the contractile ring but also in actin cortices close to the cleavage furrow, thereby masking the effects of OptoMYPT at the contractile ring. It is also possible that OptoMYPT weakened the viscoelasticity of the contractile ring, balancing with the reduction in the contractile ring. The use of a localizer that is closer to endogenous active NMII or a specific localization could result in an OptoMYPT system with better specificity and spatial resolution than the current version.

The OptoMYPT system will provide opportunities not only to understand the mechanics of morphogenesis, but also to shape the morphology of cells and tissues with precision and flexibility as desired. Recent papers have applied optogenetic systems in vivo, and succeeded in inducing arbitrary forms of apical constriction[61,62]. By combining red light-responsive optogenetic tools such as PhyB-PIF with blue light-responsive tools[63,64], it will be possible to create more sophisticated morphology with an increase or decrease in contractile force in the same cells and tissues.

## Methods

***Xenopus* embryo manipulation and microinjection**. All experiments using *Xenopus laevis* were approved by The Institutional Animal Care and Use Committee, National Institutes of Natural Sciences (Permit Number 18A038, 19A062, 20A053). *X. laevis* adults were purchased from Watanabe Zoushoku and maintained in the Division of Molecular and Developmental Biology, NIBB. Females roughly range from 2 to 8 years old and males roughly range from 1 to 3 years old. Manipulation of *X. laevis* embryos and microinjection were performed according to the following standard methods. Unfertilized eggs were obtained from female frogs injected with gonadotropin (ASKA Animal Health, Japan) and were artificially fertilized using testicular homogenates. 30 min after fertilization, eggs were de-jellied in 4% L-cysteine solution (pH adjusted to 7.8 with NaOH) and incubated at 14 °C or 17 °C in 1/10x Steinberg's solution. Plasmids for mRNA synthesis were constructed using pCSf107mT, which contains 4 x SP6/T7 transcription terminator sequences[65]. An mMachine SP6 kit (Invitrogen) was used to transcribe synthetic mRNAs from plasmid DNAs. Each four-cell stage embryo was microinjected with 50 pg of SspB-mScarlet-I or SspB-mScarlet-I-MYPT169 mRNA; 450 pg of Stargazin-mEGFP-iLID or mEGFP-KRasCT mRNA; and 150 pg of Lifeact-miRFP703 mRNA. Gastrula embryos (st. 12) were observed directly using the IX83 inverted microscope (see "Live-cell fluorescence imaging"). An oil immersion objective lens (UPLXAPO ×40, N.A. 1.4; Olympus) was used for three-dimensional imaging. The vitelline membrane was manually removed before imaging to clearly visualize the cell–cell junctions. For optogenetic experiments, blue light was illuminated from the objective lens for 500 msec every 20 s.

**Plasmids**. The oligonucleotides for PCR and DNA sequencing were purchased from FASMAC. The cDNAs of human MYPT1 and PP1c were derived from HeLa cells (Human Science Research Resources Bank). SspB-mScarlet-I-PP1BDs were obtained by Gibson assembly cloning, combining the SspB obtained from pLL7.0: tgRFPt-SSPB WT (plasmid #60415: Addgene) and cDNAs of PP1BDs. Stargazin-mEGFP-iLID was obtained by Gibson assembly cloning, combining the Stargazin derived from Stargazin-GFP-LOVpep (plasmid #80406: Addgene) and iLID derived from pLL7.0: Venus-iLID-CAAX (from KRas4B) (plasmid #60411: Addgene). CRY2 and CIBN-EGFP-KRasCT were obtained from pCX4puro-CRY2-CRaf and pCX4neo-CIBN-EGFP-KRasCT[66] and inserted into the pCAGGS vector[67]. The cDNAs of Lifeact and NES were obtained by oligo DNA annealing and ligation, and inserted into each vector. The cDNA of hyPBase, an improved PiggyBac transposase[68], was synthesized (FASMAC), and inserted into the pCAGGS vector. Tol2-based transposon donor vector (pT2A) and Tol2 transposase expression vector (pCAGGS-T2TP) were the kind gifts of Koichi Kawakami (National Institute of Genetics). The cDNA of Stargazin-mEGFP-iLID was inserted into the pT2Apuro vector (pT2Apuro-Stargazin-mEGFP-iLID). The cDNAs of rtTA[69] synthesized by FASMAC, tet-response element, and SspB-mScarlet-I-MYPT169 were inserted into a PiggyBac donor vector with ligation and Gibson assembly to generate pPBbsr2-rtTA2-TRE-SspB-mScarlet-I-MYPT169. For in vitro mRNA synthesis, these cDNAs were subcloned into the pCSf107mT vector. The nucleotide sequences of newly generated constructs including oligonucleotides for PCR and DNA sequences are provided in the benchling links of the Supplementary Table 1.

**Cell culture**. MDCK cells (no. RCB0995: RIKEN Bioresource Center) were maintained in minimal essential medium (MEM; 10370-021: ThermoFisher Scientific) supplemented with 10% fetal bovine serum (FBS; 172012-500 ML: Sigma), 1x Glutamax (35050-061: ThermoFisher), and 1 mM sodium pyruvate (11360070: ThermoFisher) in a 5% $CO_2$ humidified incubator at 37 °C. NIH-3T3 cells (no. RCB0150: RIKEN Bioresource Center), a kind gift from Jun-ichi Nakayama (National Institute for Basic Biology), were maintained in DMEM (Nacalai Tesque) supplemented with 10% FBS in a 5% $CO_2$ humidified incubator at 37 °C. When splitting the MDCK cells, the growth media was removed, and 3 mL PBS/1 mM EDTA solution was added to the cells and incubated for 5 min at 37 °C. Next, the PBS/1 mM EDTA solution was discarded, and 2 mL PBS/1 mM EDTA/0.25% Trypsin was added. The cells were incubated for 5 min at 37 °C. Prewarmed growth medium was added to resuspend the cells. When splitting the NIH-3T3 cells, the first treatment with 3 mL PBS/1 mM EDTA solution was omitted.

**Transfection**. Because PP1BDs of MYPT1 fused with fluorescent proteins tend to aggregate for long-term expression, most experiments were performed by transient expression. The MDCK cells were electroporated by using Nucleofector IIb (Lonza) according to the manufacturers' instructions (T-023 program) with a house-made DNA- and cell-suspension solution (4 mM KCl, 10 mM $MgCl_2$, 107 mM $Na_2HPO_4$, 13 mM $NaH_2PO_4$, 11 mM HEPES pH. 7.75). For transient expression, the plasmids were mixed according to the following ratios to achieve efficient membrane translocation by light: SspB:iLID = 1:4; and CRY2:CIB = 1:3. After electroporation, the cells were plated on 35-mm culture dishes or collagen-coated 35-mm glass-base dishes.

**Establishment of stable cell lines**. For transposon-mediated gene transfer, MDCK cells were transfected with PiggyBac or Tol2 donor vectors and PiggyBac or Tol2 transposase-expressing vectors at a ratio of 3:1. One day after transfections, cells were treated with 10 μg/mL blasticidin S (InvivoGen, San Diego, CA) or 1.0 μg/mL puromycin (InvivoGen) for selection. The bright bulk cell population was collected using a cell sorter (MA900; SONY).

**Live-cell fluorescence imaging**. Cells were imaged with an IX83 inverted microscope (Olympus, Tokyo) equipped with an sCMOS camera (Prime: Photometrics, Tucson, AZ; or ORCA-Fusion BT: Hamamatsu Photonics, Hamamatsu, Japan), a spinning disk confocal unit (CSU-W1; Yokogawa Electric Corporation, Tokyo), and diode lasers at wavelengths of 488 nm, 561 nm, and 640 nm. An oil immersion objective lens (UPLXAPO60XO, N.A. 1.42; UPLXAPO ×40, N.A. 1.4; Olympus), an air/dry objective lens (UPLXAPO40X, N.A. 0.95; UPLXAPO 20X, N.A. 0.8; Olympus) were used. The excitation laser and fluorescence filter settings were as follows: Excitation laser, 488 nm (mEGFP), 561 nm (mScarlet-I), and 640 nm (miRFP703); dichroic mirror, DM 405/488/561 dichroic mirror (mEGFP, mScarlet-I, and miRFP703); emission filters, 500–550 nm (mEGFP), 580–654 nm (mScarlet-I), and 665–705 nm (miRFP703). The microscope was controlled by MetaMorph software (ver. 7.10.3). During observation, cells were incubated with a stage incubator set to 37 °C and containing 5% $CO_2$ (STXG-IX3WX; Tokai Hit).

Due to the different kinetics depending on the optogenetic systems, the cells expressing iLID-SspB or CRY2-CIB system were illuminated with blue light for 500 msec less than every 20 s for iLID-SspB system and less than every 2 min intervals for CRY2-CIB. For global illumination of the blue light, blue LED light (450 nm) (LED-41VIS450; OptoCode Corp., Japan) was continuously illuminated from the top of the stage or pulsed blue light (488 nm) was illuminated through the objective lens. For local light illumination in the interphase cells, a digital micromirror device (Polygon 400; Mightex) mounted on the IX83 microscopic system, and pT-100 (CoolLED) were used. For local light illumination during cytokinesis, an SP8 FALCON inverted confocal laser scanning microscope (Leica) equipped with a water immersion objective lens (HC PL APO 63x/1.20 W motCORR; Leica) was used. The microscope was controlled by LAS X software (ver. 3.5.5). Local light illumination was started using the FRAP function just after chromosome segregation onset. We illuminated every 3.11 s, and acquired images every 15.54 s. The positions of regions of interest (ROIs) were manually corrected every 2 min in all samples.

For all time-lapse imaging, MDCK cells were plated on 35 mm glass-bottom dishes (IWAKI) or 4-well glass-bottom dishes (The Greiner Bio-One). Before time-lapse imaging, the medium was replaced with FluoroBrite (Invitrogen) supplemented with 10% FBS, 1x Glutamax.

**Immunofluorescence**. Cells were fixed with 3.7% formaldehyde in PBS for 20 min, followed by permeabilization by 5 min incubation in 0.05% Triton X-100-containing PBS. Samples were soaked for 30 min in Can Get Signal immunostain (solution A) (Toyobo, Japan) and then incubated with primary antibodies, phospho-MLC antibody (1:50 dilution; Cell Signaling Technology #3674), or PPP1CB antibody (1:200 dilution; abcam #ab53315) diluted in Can Get Signal immunostain (solution A) for 1 h at room temperature. Next, the cells were washed three times with PBS, and then incubated for 1 h at room temperature with Alexa 633-conjugated anti-rabbit IgG (1:1000 dilution; ThermoFisher) in Can Get Signal immunostain (solution A). Finally, the cells were washed three times with PBS and subjected to fluorescence imaging.

In Fig. 2a, b, the cells expressing control proteins or OptoMYPT were seeded on the four-well glass-bottom dish 2 h before light illumination. Lamellipodial regions of the cells were locally illuminated with blue light for 30 min under the microscope, and then fixed with 3.7% formaldehyde in PBS on the stage of the microscope. Subsequent immunostaining steps were also performed on the stage of the microscope. For one experiment, the locations of 10 sites per well where the cells expressed control and OptoMYPT proteins were recorded, and fluorescence images were acquired at the same positions after staining.

**Traction force microscopy**. Polyacrylamide gel substrates were prepared in accordance with previously published protocols[70,71]. In brief, the gel solution was prepared with 4% acrylamide, 0.1% bisacrylamide, 0.8% ammonium persulfate, 0.08% TEMED (Nacalai Tesque), and 5% deep red fluorescent carboxylate-modified beads (0.2 μm diameter; F8810; Thermo Fisher Scientific). In all, 13 μL of the mixture was added to a 35 mm glass-base dish (IWAKI) and then covered with a glass coverslip of 15 mm diameter (Matsunami). After gel polymerization at room temperature, the surface was coated with 0.3 mg/mL type I collagen (Nitta Gelatin, Osaka, Japan) using 4 mM sulphosuccinimidyl-6-(4-azido-2-nitrophenylamino) hexanoate (Sulfo-SANPAH; Pierce). Cells were seeded on the gel, and imaged with a spinning disk confocal microscope. To quantify the traction force, two Fiji/ImageJ (ver. 2.1.0/1.53c) plugins, i.e., the iterative PIV and FTTC plugins, were used. Note that Young's modulus of the gel was estimated as ~2 kPa according to a previous report[72]. The traction force in locally illuminated areas was used for the quantification.

**Western blotting**. Cells were lysed in 1x SDS sample buffer. After sonication, the samples were separated by 5–20% gradient SDS-polyacrylamide gel electrophoresis (Nagaiki precast gels; Oriental Instruments, Ltd.) and transferred to polyvinylidene

difluoride membranes (Millipore). After blocking with Odyssey Blocking Buffer-TBS (LICOR Biosciences) for 1 h, the membranes were incubated with primary antibodies overnight at 4 °C, followed by the secondary antibodies for 1 h at room temperature. For primary antibodies, phospho-MLC antibody (1:500 dilution; Cell Signaling Technology #3674), phospho-Ezrin/Radixin/Moesin antibody (1:500 dilution; Cell Signaling Technology #3726), MYPT1 antibody (1:500 dilution; Cell Signaling Technology #2634) and α-Tubulin antibody (DM1A) (1:5000 dilution; sc-32293: Santa Cruz Biotechnology) were diluted in Odyssey Blocking Buffer-TBS. For secondary antibodies, IRDye680LT-conjugated goat polyclonal anti-rabbit IgG (H + L) (1:5000 dilution; LI-COR Bioscience) and IRDye800CW-conjugated donkey polyclonal anti-mouse IgG (H + L) (1:5000 dilution; LI-COR Bioscience) were diluted in Odyssey Blocking Buffer-TBS. Proteins were detected with an Odyssey infrared scanner (LI-COR Bioscience).

**Laser ablation**. Laser ablation was conducted as described previously[73], using an IX81 inverted microscope (Olympus) equipped with a spinning disk confocal unit (CSU-X1; Yokogawa Electric Corporation) and iXon3 897 EM-CCD camera (Andor), and controlled with IQ2 software (Andor). A dry objective lens (UPlanAPO ×20, N.A. 0.7; Olympus) was used. An N2 Micropoint laser (16 Hz, 365 nm, 0.3 μW; Photonic Instruments) was focused on the plasma membrane at a cell–cell junction labeled with Stargazin-mEGFP-iLID or mEGFP-KRasCT. Time lapse images were acquired every 3.08 s before and after the course of the laser ablation. Before ablation, the embryos in the incubator were illuminated from overhead by blue light for 20 min. The vitelline membrane was manually removed before imaging to clearly visualize the cell–cell junctions.

**Imaging analysis**. All fluorescence imaging data were analyzed and quantified by Fiji (Image J). For all images, the background was subtracted and images were registered by StackReg, a Fiji plugin to correct misregistration, if needed. To quantify the cytoplasmic fluorescence intensity changes in Fig. 1, the ROI was selected in each image and normalized by the mean fluorescence intensity of the first 10 images under dark conditions. In Fig. 2, to quantify the fluorescence intensity ratio of the light area to dark area, we used the cytoplasmic or lamellipodial region, avoiding extremely bright regions such as stress fibers. To quantify the area of membrane protrusion in Fig. 2, the ROI was chosen so as to coincide with the local light-irradiated area. In OptoMYPT-dark cells, the ROI was a region of lamellipodia similar to that of light-illuminated cells. Fluorescence images of Lifeact-miRFP703 were binarized and the difference between the area at each time point and at $t = 0$ was calculated.

**Physical modeling**. According to previous modeling efforts on the force balance in a dividing cell[37,38,49], the temporal change in radius of the contractile ring, $R_r$, is described as

$$\alpha \frac{dR_r}{dt} = -(T_r - 2R_r T_c \cos\theta) \qquad (1)$$

where $\alpha$ is ring viscosity, $T_r$ is the tension of the contractile ring, $T_c$ is the tension in the actomyosin cortex, and $\theta$ is the angle between the equatorial plane and polar surface at the furrow (see Supplementary Fig. 8 for the details). Note that our model was simplified from Eq. S1 in the previous report[49], since we consider symmetric pole shapes. For brevity, let $F_r$ and $F_c$ denote $T_r$ (ring tension) and $2R_r T_c \cos\theta$ (cortical tension), respectively (Supplementary Fig. 8). Then, the furrow ingression rate, $v$, can be expressed as

$$v \propto F_r - F_c \qquad (2)$$

**Statistics and reproducibility**. In all box-and-whisker plots, the box shows the quartiles of data with the whiskers denoting the minimum and maximum except for the outliers detected by 1.5 times the interquartile range. All statistical analyses were conducted in Microsoft Excel software (ver. 16.54) for two-tailed Student's $t$ test (Figs. 3g, j and 4d, g and Supplementary Figs. 4b and 6e) or python (ver. 3.7) with scipy (ver. 1.4.1) (scipy.stats.brunnermunzel) for two-tailed Brunner–Munzel test (Fig. 2c). The experiments were repeated at least two times independently with similar results. The number of samples for quantifications is listed in the figure legends. For western blotting, the original images were shown in Supplementary Fig. 10.

**Reporting summary**. Further information on research design is available in the Nature Research Reporting Summary linked to this article.

## Data availability

Source data for figures and supplementary information are provided with the paper. Full western blot images are in Supplementary Fig. 10. Imaging data generated in this study have been deposited in the RIKEN SSBD:repository (Systems Science of Biological Dynamics repository) with the https://doi.org/10.24631/ssbd.repos.2021.11.002. The plasmids will be available from Addgene at https://www.addgene.org/Kazuhiro_Aoki/. The nucleotide sequences of generated constructs including oligonucleotides for PCR and DNA sequences are provided in Supplementary Table 1. Source data are provided with this paper.

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

## Acknowledgements
We thank all members of the Aoki Laboratory for their helpful discussions and assistance. We also thank Michiyuki Matsuda, Naoya Hino (Kyoto University), and Hisayo Fukuda (Kansai Medical University) for the advice on electroporation. We also thank M. Ebisuya and M. Takahashi for the comments on the manuscript. This work was supported by JSPS KAKENHI (Grant number 19J20538 to K.Y.; 19K16207, 19H05675 to Y.K.; and 18H02444, 19H05798 to K.A.), JST, CREST (Grant number JPMJCR1654 to K.A.), and Joint Research of the Exploratory Research Center on Life and Living Systems (ExCELLS) (ExCELLS program No.18-204, 19-205, 20-204 to S.S.).

## Author contributions
K.Y. and K.A. designed the research. K.Y., H.M., M.I., Y.M., and N.K., performed experiments. K.Y. and Y.K. developed the theoretical workflow. K.Y. and H.M. analyzed data. K.Y., Y.K., and K.A. wrote the manuscript, with critical input from S.T., N.U., and S.S.

## Competing interests
The authors declare no competing interests.
