## [Peer Review File · Nature Communications]

Reviewers' Comments:

Reviewer #1:

Remarks to the Author:

In this manuscript Yamamoto et al. report on the development of a new optogenetic tool, termed opto-MYPT, to locally reduce actomyosin contractility in cells, through dephosphorylation (inactivation) of non muscle myosin II (NM II) . The authors use this new tool to assess the contribution of actomyosin based cortical tension to cleavage furrow ingression during cytokinesis in MDCK cells. They find that reducing cortical tension at the poles of cells during cytokinesis increases the rate of furrow ingression.

This is the first optogenetic tool reported to reduce local actomyosin contractility, and therefore of possible interest for the biosciences community. Overall the manuscript is properly written and structured, and the figures are well presented and clear.

However, the authors should improve on the validation of this new tool, as some of the conclusions are not sufficiently supported by the data presented in this manuscript. My main concern is that not much evidence is provided that this tool efficiently and specifically induces dephosphorylation of NM II. Furthermore, the evidence that local recruitment of opto-MYPT results in reduced contractility is somewhat limited.

For this reason, I cannot yet support publication of this manuscript in Nature Communications.

Main concerns:

Validation of opto-MYPT:

MYPT is one of the 3 subunits of the Myosin Light Chain Phosphatase (MLCP) complex which dephosphorylates (inactivates) non muscle myosin II (NMII), which results in a reduction in actomyosin contractility. Full length MYPT binds to the catalytic subunit PP1c (and increases its specificity for myosin), through its PP1BD(located in aa 1-296), and it binds directly to myosin via a C-terminally located MHC binding domain. MYPT controls both targeting, specificity and activity of the phosphatase subunit, which also has hundreds of other targets in the cell.

Opto-MYPT comprises a portion of the PP1BD (MYPT169) fused to one half of the optogenetic dimerizers ILID/nano or Cry2/CIB, and the other half targeted to the plasma membrane(PM). This allows light induced recruitment of the PP1BD to the PM. This optogenetic relocation strategy works very well as shown in figure 1, and is in line with reports from various other groups that have successfully used ILID or Cry2-CIB to recruit proteins to the plasma membrane (also discussed here: <https://doi.org/10.1016/j.ceb.2020.03.003>, and here DOI: 10.1242/dev.175067) The authors show convincingly that the catalytic phosphatase subunit PP1c is recruited along with the MYPT169 fusion protein(although the authors only show this for overexpressed PP1c). It would be nice however to see what happens with endogenous PP1c. PP1c has hundreds of targets, and I wonder if light induced translocation of MYPT169 might sequester the complete pool of endogenous PP1c.

One big concern here is that over-expression of opto-MYPT, could have a dominant effect on the function of PP1c. E.g. It has been demonstrated previously that PP1c in complex with MYPT, increases its catalytic activity ~15 fold. DOI: 10.1038/nature02582

Additionally, it could be that having a ton of the MYPT169 fusion protein in the cytosol, competes with the endogenous MLCP complex, and it could actually hamper dephosphorylation of MLC, thereby increasing contractility.

The authors don't really quantify how the expression of opto MYPT in their stable cell lines compares to endogenous MYPT levels, but looking at the microscopy data it appears quite high.

And it indeed looks as if expression of MYPT fragments has a dominant effect on the localization of PP1c. E.g. in Figure 1 D) In control cells PP1c-miRFP703 localizes to the nucleus, but in the MYPT expressing cells it seems that there's no PP1c in the nucleus. Is this because the MYPT truncations sequester the protein in the cytoplasm? This is something that might be important to discuss.

Furthermore, there seems to be a big difference in cell size between control and MYPT expressing cells in figure 1D. The control cells appear much smaller and less flat. (although, in supplementary figure S1A this is not as clear). Do these MYPT fragments hyperactivate PP1c?

Another concern is that MYPT169 lacks myosin interacting domains, and therefore the question is how efficient the MYPT169/PP1c complex is at specifically dephosphorylating myosin, by recruiting it to the PM (is the catalytic subunit brought close enough to myosin via this strategy?), and how specific, considering that PP1c has so many targets?

Does Opto-MYPT cause dephosphorylation of ppMLC in response to a light stimulus?

The evidence that opto-MYPT actually causes dephosphorylation of MLC in response to a local light stimulus is a bit limited. The only evidence provided is an immunofluorescence experiment, in which the authors locally illuminate cells (control cells or cells expressing opto-MYPT), and fix and immunostain for phosphorylated MLC. The authors observe a small but significant difference. The images presented in figure 2b are not very convincing. The illuminated area (blue circle) in the control cell is at the edge of the cell cluster, while in the opto-MYPT cell it is in an area where the cell contacts another cell. It makes it very hard to assess if the decreased fluorescence is due to the optogenetic stimulus.. or simply because of the location.. (BTW. it is also unclear how the authors performed this experiment.. How did they find the cells that were locally illuminated after fixing and staining? Was the fixation procedure and immunostaining performed on the microscope, and one cell at a time?)

Also: wouldn't it be easier to illuminate the whole well, and look at overall ppMLC levels, instead of locally, because the ppMLC signal varies so much throughout the cell. (if needed the authors could look specifically at the myosin near the membrane by using TIRF. And maybe the authors could take along the MYPT 296 fragment as a positive control, as this one seemed to completely abolish phosphorylation of MLC(suppl fig S1C).

Another important experiment that I missed is a western blot to test if ppMLC is decreased after blue light illumination.

Does opto-MYPT reduce actomyosin contractility?

The authors immediately went for quite a difficult experiment to test if opto-MYPT recruitment to the PM reduces contractility, namely traction force microscopy. And they find a small decrease in traction forces upon blue light stimulation. However, a control experiment is missing with cells not expressing the MYPT containing half. It is also unclear to me why the authors choose to perform local illumination experiments. Why not start out with illuminating the whole cell?

Wouldn't it have been better to use a different method to study contractility? For example, by looking at cell junction length (similar to what was done by Cavanaugh et al 2020, <https://www.sciencedirect.com/science/article/pii/S1534580719310238?via=ihub>). MDCK cells should be ideally suited for such experiments.

Or alternatively to look at the dissolution of stressfibers in light stimulated cells? The cell type used by the authors doesn't show many stressfibers, but perhaps they could use 3T3 fibroblasts..

Overall I missed experiments where the authors look more directly at the actin cytoskeleton and effects of opto-MYPT recruitment on actin organization.

About the cytokinesis experiments in figure 4 I have a similar comment.

Why didn't the authors do an experiment where the authors illuminate the central portion of the anaphase spindle. This should inhibit/ slow down cleavage furrow ingression, and would give the most clear readout for a reduction in actomyosin contractility. This would not only serve to show that the opto-MYPT works well, but it would also demonstrate spatial control more clearly. In the pole illumination experiments, one can actually see recruitment of opto-MYPT along most of the cortex, and not just the poles(although it appears that there's a bit less near the cleavage furrow). Such experiments would really help to validate that this tool works as expected.

About the methodology used. The authors do not provide detailed information about how the optogenetic stimulation was performed. Did they use pulsed or continuous illumination? What was the pulse frequency, Radiant exposure per pulse, or the irradiance?

Minor concerns:

Abstract:

Line 33: "The system, named optoMYPT, combines a catalytic subunit of the type 1 phosphatase-binding domain of MYPT with an optogenetic dimerizer".

Catalytic subunit seems incorrect here. It is the PP1c-binding domain of MYPT, so more like an adapter domain that subsequently binds the phosphatase. Or does binding of the PP1BD to the phosphatase increase phosphatase activity?

Figure 3D) I cannot distinguish the colors of the individual cells of the control data and the Opto-MYPT-dark data.

Reviewer #2:

Remarks to the Author:
Review report Aoki, 2021 Nat Com.

In this manuscript, Yamamoto and colleagues describe the development and characterisation of a new optogenetic tool, OptoMYPT, that can be used to locally inactivate non-muscle Myosin-II at the plasma membrane (PM). They test different variants in their ability to be recruited to the PM and deactivate Myosin-II locally. Using the tool, they show that local deactivation of Myosin-II causes (1) membrane protrusion in MDCK cells, and (2) increases the ingress rate of cleavage furrows during cytokinesis.

The manuscript is technically sound and clearly and concisely written. To my knowledge, this is the first available optogenetic tool to locally decrease contractility in live samples. This is a major step forward in the field and useful for many researchers in cytoskeleton biology.

We believe the paper could be accepted after some revisions, described below:

- 1) Western blot in S1C. The lane of the MYPT296 sample has much lower tubulin expression compared to the other samples. Quantification or more uniform loading is necessary to evaluate the effects of the expression of MYPT296 on ppMLC.
- 2) The authors showed by western blot the basal ppMLC after the expression of the different optogenetic construct. If technically feasible in their lab, the authors could consider to show the decrease of ppMLC by western blot after illuminating the entire well with blue light.
- 3) Movie S3, directly after light stimulation it seems that there is a transient increase in contraction. Why? Is it something repeatable?
- 4) The effects of the optoMYPT stimulation are quite small in some of the assays (e.g. 2e,f), maybe the authors should consider to include a high basal contractile state in their assays (fibronectin coating to get good contractility). This could potentially show the dynamic range of the activated tool better.
- 5) Figure 4g, Why does OptoMYPT-dark speed up the ingress rate? Is the optogenetic MYPT system leaky in this context? The authors should investigate this further, and perhaps elaborate on expression levels. It is important to characterize the effects of optoMYPT stimulation from the basal overexpression effect.

Reviewer #3:

Remarks to the Author:

In this manuscript, the authors develop an optogenetic controller of MYPT1 to recruit with high temporal and spatial resolution protein phosphatase 1 to the cortex so that it can reduce the phosphorylation of myosin II light chain. By dephosphorylating MLC, myosin II is presumed to have lower activity in that region, leading to a reduction of myosin II-mediated force production. The authors then use their system to attempt to discern the myosin II contribution to tension in various locations around the cell, including during cytokinesis.

Overall, the strength of this paper is the potential of the opto-MYPT1 controller that the authors have developed. They do a reasonable job testing the effects of the controller. However, the major biological insights in this paper are fairly small given the history of deciphering the mechanics of cytokinesis. This is fine for this paper, as I would view this paper as more of a tool development study. Some questions do persist regarding what the authors are actually examining that come up throughout the paper.

Specific comments/questions:

1. The authors suggest that by recruiting PP1c that they are then specifically reducing phosphorylation of myosin II. But, PP1c appears to be fairly promiscuous and there are other known substrates such as moesin and adducin. What other substrates can their opto-MYPT1(169) control promote dephosphorylation of if all that is happening is PP1c is being recruited to the

cortex, especially since the MLC binding site has been removed?

2. They do look at a pan-phospho-ERM in Fig. S1, but oddly this is never described in the Results section. I am not sure that pan-phospho-ERM is specific enough however. Two points here: this result needs to be pointed out in the Results, but with more specific probing should be provided if possible. What is the relative abundance of the different ERMs? Even if moesin is much less abundant, that does not mean that its impact is just as significant (relative abundance alone is not a sole indicator of contribution).

3. Fig. 2C, given the nearly bimodal nature of the control group, a Student t-test would not be the appropriate statistical test. If one considered a non-normal distribution, are these data still considered statistically significant?

4. Critical controls seem to be missing. It would be useful to see movies of the controls for the traction force experiment and the protrusion formation corresponding to movies S3 and S4, respectively. The effects provided are not overwhelming. Similar quantification of controls should be provided for Fig. 2f. The authors do provide an example image in Fig. 3 of control cells along with quantification, but again it would be useful to see example movies.

5. The premise about the controversy over the relative contributions of cortical tension to cytokinesis is extremely selective and overlooks considerable amount of research on this topic. This text needs to be significantly modernized.

6. Bleb formation results from the balance between cortical tension and membrane-cortex anchoring density so simply counting blebs as an indicator of cortical tension might be correct, but it might not be. Also, as mentioned above, other proteins, such as moesin, that help mediate membrane-cortex anchoring, are also reported as potential substrates of PP1c. Also, some of the anchoring proteins require myosin II to tug on them to get them to lock in. Further, in the Opto-MYPT1(169) scenario where you are still getting blebs, but bigger ones, this could be because the membrane-cortex linkages are even weaker. These effects are complicated for sure. Therefore, I think it is better to just be descriptive in what the changes in cell phenotype are, rather than jumping to the conclusion that cortical tension is all that is getting affected.

7. I agree that changes in the rates of furrow ingression can reflect differences cortical tension, but myosin II can also impact strain stiffening of the network, which similarly can impact furrow ingression rates. In fact, both effects are contributors to furrow ingression rates. I would simplify this portion of the discussion without solely implicating one parameter vs. the other – better to be inclusive of potential mechanisms.

It should be appreciated that the authors do point out many of these issues described above in the Discussion, which I was happy to see. I think they need to go back through Results and present each point similarly using qualified language so that they do not over-interpret their observations from the outset.

General Statements

We thank the reviewers for their critical comments and valuable suggestions, which have helped us to improve our paper. Our point-by-point responses are given below, with the comments of the reviewers shown in *dark blue italics*. As detailed in our responses, we revised and expanded numerous explanatory passages in the main text.

REVIEWER COMMENTS

Reviewer #1 (Remarks to the Author):

In this manuscript Yamamoto et al. report on the development of a new optogenetic tool, termed opto-MYPT, to locally reduce actomyosin contractility in cells, through dephosphorylation (inactivation) of non muscle myosin II (NM II) . The authors use this new tool to assess the contribution of actomyosin based cortical tension to cleavage furrow ingression during cytokinesis in MDCK cells. They find that reducing cortical tension at the poles of cells during cytokinesis increases the rate of furrow ingression.

This is the first optogenetic tool reported to reduce local actomyosin contractility, and therefore of possible interest for the biosciences community. Overall the manuscript is properly written and structured, and the figures are well presented and clear.

However, the authors should improve on the validation of this new tool, as some of the conclusions are not sufficiently supported by the data presented in this manuscript. My main concern is that not much evidence is provided that this tool efficiently and specifically induces dephosphorylation of NM II. Furthermore, the evidence that local recruitment of opto-MYPT results in reduced contractility is somewhat limited.

For this reason, I cannot yet support publication of this manuscript in Nature Communications.

We would like to thank the reviewer for the critical comments.

Main concerns:

Reviewer 1-Question 1 (R1-Q1): *Validation of opto-MYPT:*

MYPT is one of the 3 subunits of the Myosin Light Chain Phosphatase (MLCP) complex which dephosphorylates (inactivates) non muscle myosin II (NMII), which results in a reduction in actomyosin contractility. Full length MYPT binds to the catalytic subunit PP1c (and increases its specificity for myosin), through its PP1BD(located in aa 1-296), and it binds directly to myosin via a C-terminally located MHC binding domain. MYPT controls both targeting, specificity and activity of the phosphatase subunit, which also has hundreds of other targets in the cell.

Opto-MYPT comprises a portion of the PP1BD (MYPT169) fused to one half of the optogenetic dimerizers ILID/nano or Cry2/CIB, and the other half targeted to the plasma membrane(PM). This allows light induced recruitment of the PP1BD to the PM. This optogenetic relocalization strategy works very well as shown in figure 1, and is in line with reports from various other groups that have

successfully used ILID or Cry2-CIB to recruit proteins to the plasma membrane (also discussed here: <https://doi.org/10.1016/j.ceb.2020.03.003>, and here DOI: 10.1242/dev.175067).

The authors show convincingly that the catalytic phosphatase subunit PP1c is recruited along with the MYPT169 fusion protein (although the authors only show this for overexpressed PP1c). It would be nice however to see what happens with endogenous PP1c. PP1c has hundreds of targets, and I wonder if light induced translocation of MYPT169 might sequester the complete pool of endogenous PP1c.

One big concern here is that over-expression of opto-MYPT, could have a dominant effect on the function of PP1c. E.g. It has been demonstrated previously that PP1c in complex with MYPT, increases its catalytic activity ~15 fold. DOI: 10.1038/nature02582.

Additionally, it could be that having a ton of the MYPT169 fusion protein in the cytosol, competes with the endogenous MLCP complex, and it could actually hamper dephosphorylation of MLC, thereby increasing contractility.

Reviewer 1-Answer 1 (R1-A1): We agree with the reviewer's concerns. Regarding the endogenous PP1c localization, we confirmed that OptoMYPT induced translocation of PP1c from the cytoplasm to plasma membrane on blue light illumination by immunostaining with anti-PP1c antibody. We have included this data in the revised manuscript (page 5, line 142, Fig. S1c).

We examined the effect of overexpression of OptoMYPT on MLC phosphorylation. As shown in Figure S1a and S1e, the expression of SspB-mScarlet-I-MYPT296 resulted in aberrant morphology with elongated protrusions in MDCK cells and the dephosphorylation of MLC, suggesting that the expression of MYPT296 elevated the phosphatase activity of endogenous PP1c. The stable expression of SspB-mScarlet-I-MYPT169, which was used as OptoMYPT, did not have any detectable effect on cellular morphology and basal MLC phosphorylation (Fig. S1a and S1e). However, we noticed that the Dox-inducible (Fig. S4) and transient expression (Figs. 4g and 5b) of SspB-mScarlet-I-MYPT169 might have an influence on basal MLC phosphorylation. For these reasons, we added the following notes to the revised manuscript:

(Page 7, line 206)

"Note that just OptoMYPT expression induced a slight decrease in phosphorylated MLC levels even before blue light irradiation (Supplementary Fig. 4), implying that PP1c is partially activated by overexpression of MYPT169 with the doxycycline-inducible expression system."

(Page 14, line 401)

"Of note, the ingress rate of OptoMYPT-dark cells was slightly higher than that of Control-pole cells (Fig. 4g), suggesting the basal effects of OptoMYPT on MLC phosphorylation even under the dark condition."

(Page 16, line 453)

"OptoMYPT-dark seems to have a lower percentage of oscillation than Control-pole (Fig. 5b), possibly due to the decrease in basal cortical tension by the expression of OptoMYPT."

R1-Q2: *The authors don't really quantify how the expression of opto MYPT in their stable cell lines compares to endogenous MYPT levels, but looking at the microscopy data it appears quite high. And it indeed looks as if expression of MYPT fragments has a dominant effect on the localization of PP1c. E.g. in Figure 1 D) In control cells PP1c-miRFP703 localizes to the nucleus, but in the MYPT expressing cells it seems that there's no PP1c in the nucleus. Is this because the MYPT truncations sequester the protein in the cytoplasm? This is something that might be important to discuss.*

R1-A2: We quantified the relative expression level of OptoMYPT compared with endogenous MYPT. Our rough estimation demonstrates that the stable- and transient-expression experiments showed approximately 12-fold and 50-fold increase relative to the endogenous MYPT level, respectively (Fig. S3).

As the reviewer suggested, PP1c is localized at the cytoplasm and nucleus (Fig. 1d and Fig. S1c), and the nuclear localized PP1c is diminished by the expression of OptoMYPT proteins (Fig. S1c). This could be because OptoMYPT proteins, which were fused with NES, sequester PP1c proteins in the cytoplasm. There may be proteins that are dephosphorylated by PP1c in the nucleus, and therefore the nuclear exclusion of PP1c by OptoMYPT may disturb its nuclear function. We have included this important note in the Discussion of the revised manuscript as follows:

(page 19, line 531)

“In addition, the expression of PP1BD of MYPT affected the localization of endogenous PP1c (Fig. 1d and Supplementary Fig. 1c). Some proteins are dephosphorylated by PP1c in the nucleus (Kiss et al. 2008), and therefore the nuclear exclusion of PP1c by OptoMYPT may disturb its nuclear function.”

R1-Q3: *Furthermore, there seems to be a big difference in cell size between control and MYPT expressing cells in figure 1D. The control cells appear much smaller and less flat. (although, in supplementary figure S1A this is not as clear). Do these MYPT fragments hyperactivate PP1c?*

R1-A3: The difference between Figure 1d and Figure S1a could be due to the difference of seeding cell density; i.e., there was a higher density in Figure 1d than in Figure S1a. We quantified the cell size in Figure 1d, and found no clear differences between the control and MYPT-expressing cells (Fig. S1d). In regard to the possibility of hyperactivation of PP1c by the expression of MYPT fragments, please see our comment above (**R1-A1**).

R1-Q4: *Another concern is that MYPT169 lacks myosin interacting domains, and therefore the question is how efficient the MYPT169/PP1c complex is at specifically dephosphorylating myosin, by recruiting it to the PM (is the catalytic subunit brought close enough to myosin via this strategy?), and how specific, considering that PP1c has so many targets?*

R1-A4: As mentioned above, MYPT296, which includes a myosin-interacting domain, showed higher basal activity of PP1c, and therefore it was not suitable for OptoMYPT. At this moment, we have not fully addressed the issue of specificity, but our data indicated, at least in part, the specificity that OptoMYPT did not dephosphorylate ERM proteins (Fig. S1e). We have carefully described the

concerns in regard to specificity and potential limitations of the experiments in the Discussion of the revised manuscript as follow:

(page 19, line 529)

“There still remain some issues to be addressed with respect to the OptoMYPT. First is the issue of substrate specificity in OptoMYPT. We could not exclude the possibility that the OptoMYPT dephosphorylates additional substrates other than MLC. In addition, the expression of PP1BD of MYPT affected the localization of endogenous PP1c (Fig. 1d and Supplementary Fig. 1c). Some proteins are dephosphorylated by PP1c in the nucleus (Kiss et al. 2008), and therefore the nuclear exclusion of PP1c by OptoMYPT may disturb its nuclear function. However, based on the fact that the expression of MYPT296 did not dephosphorylate ERM proteins (Supplementary Fig. 1e), it seems somewhat unlikely that OptoMYPT dephosphorylates and inactivates the ERM proteins.”

R1-Q5: *Does Opto-MYPT cause dephosphorylation of ppMLC in response to a light stimulus? The evidence that opto-MYPT actually causes dephosphorylation of MLC in response to a local light stimulus is a bit limited. The only evidence provided is an immunofluorescence experiment, in which the authors locally illuminate cells (control cells or cells expressing opto-MYPT), and fix and immunostain for phosphorylated MLC. The authors observe a small but significant difference. The images presented in figure 2b are not very convincing. The illuminated area (blue circle) in the control cell is at the edge of the cell cluster, while in the opto-MYPT cell it is in an area where the cell contacts another cell. It makes it very hard to assess if the decreased fluorescence is due to the optogenetic stimulus.. or simply because of the location.. (BTW. it is also unclear how the authors performed this experiment.. How did they find the cells that were locally illuminated after fixing and staining? Was the fixation procedure and immunostaining performed on the microscope, and one cell at a time?)*

R1-A5: We apologize for creating confusion in regard to the representation of MLC dephosphorylation induced by OptoMYPT. To clarify this subject, we obtained new data in sparsely seeded cells and replaced the original images with new images (Fig. 2b, 2c). Further, In the revised manuscript, we described the procedures for OptoMYPT with ppMLC immunostaining experiments in greater detail as follows:

(page 22, line 652)

“In Figure 2a and 2b, the cells expressing control proteins or OptoMYPT were seeded on the 4-well glass-bottom dish 2 h before light illumination to establish the sparsely migrating conditions. Lamellipodial regions of the cells were locally illuminated with blue light for 30 min under the microscope, and then fixed with 3.7% formaldehyde in PBS on the stage of the microscope. Subsequent immunostaining steps were also performed on the stage of the microscope. For one experiment, the locations of 10 sites per well where the cells expressed control and OptoMYPT proteins were recorded, and fluorescence images were acquired at the same positions after staining.”

R1-Q6: *Also: wouldn't it be easier to illuminate the whole well, and look at overall ppMLC levels, instead of locally, because the ppMLC signal varies so much throughout the cell. (if needed the*

authors could look specifically at the myosin near the membrane by using TIRF. And maybe the authors could take along the MYPT 296 fragment as a positive control, as this one seemed to completely abolish phosphorylation of MLC(suppl fig S1C).

Another important experiment that I missed is a western blot to test if ppMLC is decreased after blue light illumination.

R1-A6: According to the reviewer's suggestion, we performed western blot experiments. As expected, ppMLC was partially decreased after blue light illumination (Fig. S4). The reduction of ppMLC levels was partial, and this was probably attributable to the characteristic design of OptoMYPT; the OptoMYPT dephosphorylates only the MLCs located beneath the plasma membrane upon light stimulation. Thus, OptoMYPT could not access a large fraction of MLCs that were located at stress fibers and other cytoplasmic compartments. We have included the data in Figure S4. We have not done the TIRF experiments because we do not have a TIRF microscope.

R1-Q7: *Does opto-MYPT reduce actomyosin contractility?*

The authors immediately went for quite a difficult experiment to test if opto-MYPT recruitment to the PM reduces contractility, namely traction force microscopy. And they find a small decrease in traction forces upon blue light stimulation. However, a control experiment is missing with cells not expressing the MYPT containing half. It is also unclear to me why the authors choose to perform local illumination experiments. Why not start out with illuminating the whole cell?

R1-A7: First, we have included the control experiments for the analysis of traction force microscopy (Fig. 3b and 3d) in the revised manuscript. While we had not tried the traction force microscopy under whole-cell illumination for the following reason, we observed the membrane protrusions upon whole-cell illumination (Fig. S5a). However, the morphological changes seemed to be weaker than those in the local illumination experiments (Fig. 2e, h). Therefore, we started out with the traction force microscopy under local illumination, and then found that the reduction of traction force was small but significant. This result led us to believe that the whole-cell illumination would not substantially change traction forces. This is supported by the fact that the effect of OptoMYPT on stress fibers, which are associated with traction forces, is limited (Fig. S5b, S5c)(see **R1-A8** for more details).

R1-Q8: *Wouldn't it have been better to use a different method to study contractility? For example, by looking at cell junction length (similar to what was done by by Cavanaugh et al 2020, <https://www.sciencedirect.com/science/article/pii/S1534580719310238?via=ihub>). MDCK cells should be ideally suited for such experiments.*

Or alternatively to look at the dissolution of stressfibers in light stimulated cells? The cell type used by the authors doesn't show many stressfibers, but perhaps they could use 3T3 fibroblasts..

Overall I missed experiments where the authors look more directly at the actin cytoskeleton and effects of opto-MYPT recruitment on actin organization.

R1-A8: We would like to thank the reviewer for the helpful comments. We used NIH-3T3 fibroblasts to examine the effects of OptoMYPT. As in MDCK cells, local illumination of blue light induced membrane protrusions from the pre-existing lamellipodia in NIH-3T3 cells (Fig. S5b, S5c, inset 1, Movie S5), but did not induce the dissolution of stress fibers and protrusions from the membrane in

close proximity to stress fibers (Fig. S5b, S5c, inset 2, Movie S5). We have also added the following discussion in the revised manuscript:

(page 19, line 539)

“It is plausible that the OptoMYPT dephosphorylates and inactivates only the NMII existing in the vicinity of the plasma membrane, such as in the lamellipodial region (Fig. 2) and at cell-cell junctions in *Xenopus* embryos (Fig. 3). Meanwhile, our data suggest that OptoMYPT is incapable of dephosphorylating MLCs that are located away from the plasma membrane and/or that are incorporated in the highly bundled actin fibers, such as stress fibers (Supplementary Fig. 5b, 5c) and the contractile ring (Supplementary Fig. 9).”

With respect to the cell-cell junction, at first, we used MDCK cell sheets to observe the effect of OptoMYPT on the cell-cell junction, but we did not see any differences. This could be because the cell-cell junctions in MDCK cell sheets are not strongly stretched. For this reason, we decided to use *Xenopus* embryos, in which myosin heavy chains are involved in gastrulation. Four-cell stage embryos were microinjected with mRNAs for OptoMYPT, and gastrula embryos (st. 12) were observed with an inverted microscope. We found that in embryos expressing OptoMYPT constructs, the cell-cell junctions became wavy in shape by blue light illumination, suggesting a decrease of actomyosin contractility (Fig. 3e-g, Fig. S6a-c, Movie S7). To directly measure the decrease in actomyosin contractility by OptoMYPT, we performed laser ablation experiments with optogenetics. The recoil velocity of the cell-cell boundary by laser ablation was significantly slower in OptoMYPT-light cells than the control experiments (Fig. 3h-j, Fig. S6d, e, Movie S8). These data clearly indicated the reduction of tension at the cell-cell junction by OptoMYPT. We have included these results in the revised manuscript (page 10, line 291).

R1-Q9: *About the cytokinesis experiments in figure 4 I have a similar comment.*

Why didn't the authors do an experiment where the authors illuminate the central portion of the anaphase spindle. This should inhibit/ slow down cleavage furrow ingression, and would give the most clear readout for a reduction in actomyosin contractility. This would not only serve to show that the opto-MYPT works well, but it would also demonstrate spatial control more clearly. In the pole illumination experiments, one can actually see recruitment of opto-MYPT along most of the cortex, and not just the poles(although it appears that there's a bit less near the cleavage furrow). Such experiments would really help to validate that this tool works as expected.

R1-A9: According to the reviewer's suggestion, we did the local illumination at the cleavage furrow in OptoMYPT-expressing cells (Fig. S9). However, the ingression rate of the cleavage furrow did not change under our experimental conditions (Fig. S9). Although the experimental results differed from our expectations, we believe the results are important and have included them in the revised manuscript (page 16, line 454, Fig. S9) in the Result section, and along with the following sentences of explanation in the Discussion:

(page 19, line 541).

“Meanwhile, our data suggest that OptoMYPT is incapable of dephosphorylating MLCs that are located away from the plasma membrane and/or that are incorporated in the highly bundled actin fibers, such as stress fibers (Supplementary Fig. 5b, 5c) and the contractile ring

(Supplementary Fig. 9). Especially in the latter case, it is technically difficult to confine local activation of OptoMYPT at the contractile ring by conventional confocal microscopy, because of the diffusion of OptoMYPT proteins and the shape of the point spread function on the defocused planes with a high numerical aperture objective lens (Supplementary Fig. 9). These effects might lead to the dephosphorylation of MLC not only in the contractile ring but also in actin cortices close to the cleavage furrow, thereby masking the effects of OptoMYPT at the contractile ring. It is also plausible that OptoMYPT weakened the viscoelasticity of the contractile ring, resulting in a balance between actomyosin contractile force and viscoelasticity of the contractile ring.”

Meanwhile, during this revision, we conducted an experiment, in which only a single-pole was irradiated with blue light during the cytokinesis of cells expressing OptoMYPT. The results showed that cytokinetic cell shape oscillation often occurred (Fig. 5a, b). It is reasonable to think that the cortical tension on both sides is out of balance by OptoMYPT, and the imbalance induces large bleb formation at the irradiated pole, leading to the cell shape oscillation. Thus, we have included these data in the revised manuscript (Fig. 5a, b).

R1-Q10: About the methodology used. The authors do not provide detailed information about how the optogenetic stimulation was performed. Did they use pulsed or continuous illumination? What was the pulse frequency, Radiant exposure per pulse, or the irradiance?

R1-A10: We apologize for the insufficient explanation. We have added detailed information of the optogenetic stimulation in the Figure legends and Materials and Methods section of the revised manuscript.

R1-Q11: Minor concerns:

Abstract:

Line 33: “The system, named optoMYPT, combines a catalytic subunit of the type 1 phosphatase-binding domain of MYPT with an optogenetic dimerizer”.

Catalytic subunit seems incorrect here. It is the PP1c-binding domain of MYPT, so more like an adapter domain that subsequently binds the phosphatase. Or does binding of the PP1BD to the phosphatase increase phosphatase activity?

R1-A11: We have corrected this mistake.

R1-Q12: Figure 3D) I cannot distinguish the colors of the individual cells of the control data and the Opto-MYPT-dark data.

R1-A12: The graph has been divided into three panels for easy viewing.

=====
Reviewer #2 (Remarks to the Author):

Review report Aoki, 2021 Nat Com.

In this manuscript, Yamamoto and colleagues describe the development and characterisation of a new optogenetic tool, OptoMYPT, that can be used to locally inactivate non-muscle Myosin-II at the plasma membrane (PM). They test different variants in their ability to be recruited to the PM and deactivate Myosin-II locally. Using the tool, they show that local deactivation of Myosin-II causes (1) membrane protrusion in MDCK cells, and (2) increases the ingression rate of cleavage furrows during cytokinesis. The manuscript is technically sound and clearly and concisely written. To my knowledge, this is the first available optogenetic tool to locally decrease contractility in live samples. This is a major step forward in the field and useful for many researchers in cytoskeleton biology. We believe the paper could be accepted after some revisions, described below:

We would like to thank the reviewer for the support of our study.

Reviewer 2-Question 1 (R2-Q1): *1) Western blot in S1C. The lane of the MYPT296 sample has much lower tubulin expression compared to the other samples. Quantification or more uniform loading is necessary to evaluate the effects of the expression of MYPT296 on ppMLC.*

Reviewer 2-Answer 1 (R2-A1): According to the reviewer's suggestion, we have repeated the same experiments and replaced the original figure with the new data to avoid misunderstanding (Fig. S1e).

R2-Q2: *2) The authors showed by western blot the basal ppMLC after the expression of the different optogenetic construct. If technically feasible in their lab, the authors could consider to show the decrease of ppMLC by western blot after illuminating the entire well with blue light.*

R2-A2: We appreciate the reviewer's comment. ppMLC was analyzed by western blotting, showing that the ppMLC level was partially decreased after blue light illumination (Fig. S4). The reason why the reduction of ppMLC levels was partial is possibly because of the the design of OptoMYPT; the OptoMYPT dephosphorylates only the MLCs located beneath the plasma membrane upon light stimulation. Thus, OptoMYPT could not access a large fraction of MLCs that were located at stress fibers and other cytoplasmic compartments. We have included the following explanation in the Discussion of the revised manuscript:

(page 19, line 539)

"It is plausible that the OptoMYPT dephosphorylates and inactivates only the NMII existing in the vicinity of the plasma membrane, such as in the lamellipodial region (Fig. 2) and at cell-cell junctions in *Xenopus* embryos (Fig. 3). Meanwhile, our data suggest that OptoMYPT is incapable of dephosphorylating MLCs that are located away from the plasma membrane and/or that are incorporated in the highly bundled actin fibers, such as stress fibers (Supplementary Fig. 5b, 5c) and the contractile ring (Supplementary Fig. 9)."

R2-Q3: *3) Movie S3, directly after light stimulation it seems that there is a transient increase in contraction. Why? Is it something repeatable?*

R2-A3: Although the right panel of Movie S6 in the revised manuscript, which was identical to the Movie S3 in the original manuscript, exhibited a transient increase in traction force, we analyzed the other data, and found that the transient increase in contraction just after blue light illumination was not reproducible. In addition, we did not observe the transient increase in traction force in the control experiments (Fig. 3b, Movie S6, left).

R2-Q4: 4) The effects of the optoMYPT stimulation are quite small in some of the assays (e.g. 2e,f), maybe the authors should consider to include a high basal contractile state in their assays (fibronectin coating to get good contractility). This could potentially show the dynamic range of the activated tool better.

R2-A4: We would like to thank the reviewer for the helpful suggestion. For this purpose, we employed NIH-3T3 fibroblasts, in which stress fibers are evident. As in MDCK cells, local illumination of blue light induced further membrane protrusions from the pre-existing lamellipodia in NIH-3T3 cells (Fig. S5b, S5c, inset 1, Movie S5), but did not induce the change in stress fiber formation and membrane protrusions from the membrane in close proximity to stress fibers (Fig. S5b, S5c, inset 2; Movie S5). To incorporate these results, we have included the following explanation in the revised manuscript:

(page 8, line 227)

“Similarly, in NIH-3T3 cells, the local illumination of blue light induced membrane protrusions from the pre-existing lamellipodia (Supplementary Fig. 5b, 5c, inset 1, Supplementary Movie 5), but did not induce the change in stress fiber formation and membrane protrusions from the site in close proximity to stress fibers (Supplementary Fig. 5b, 5c, inset 2). This result suggests that MLCs on stress fibers probably escape the dephosphorylation by OptoMYPT because they are far from the plasma membrane. “

R2-Q5: 5) Figure 4g, Why does OptoMYPT-dark speed up the ingress rate? Is the optogenetic MYPT system leaky in this context? The authors should investigate this further, and perhaps elaborate on expression levels. It is important to characterize the effects of optoMYPT stimulation from the basal overexpression effect.

R2-A5: We appreciate the reviewer’s comments. Like the reviewer, we also considered that the expression of OptoMYPT might reduce ppMLC levels and actomyosin contractility even under dark conditions in a manner dependent on the expression level. As described in **R1-A2**, we quantified the relative expression level of OptoMYPT compared with endogenous MYPT. Our rough estimation demonstrates that the stable- and transient-expression experiments showed approximately 12-fold and 50-fold increase relative to the endogenous MYPT level, respectively (Fig. S3). In Figure S1e, the OptoMYPT (MYPT169) did not reduce basal ppMLC levels because a stable cell line was used in this experiment. Meanwhile, the Dox-inducible (Fig. S4) and transient expression (Fig. 4g and Fig. 5b) implied a leak of MYPT on basal ppMLC levels. Therefore, we have added the following notes to the revised manuscript:

(Page 7, line 206)

“Note that just OptoMYPT expression induced a slight decrease in phosphorylated MLC levels even before blue light irradiation (Supplementary Fig. 4), implying that PP1c is partially activated by overexpression of MYPT169 with the doxycycline-inducible expression system.”

(Page 14, line 401)

“Of note, the ingress rate of OptoMYPT-dark cells was slightly higher than that of Control-pole cells (Fig. 4g), suggesting the basal effects of OptoMYPT on MLC phosphorylation even under the dark condition.”

(Page 16, line 453)

“OptoMYPT-dark seems to have a lower percentage of oscillation than Control-pole (Fig. 5b), possibly due to the decrease in basal cortical tension by the expression of OptoMYPT.”

Reviewer #3 (Remarks to the Author):

In this manuscript, the authors develop an optogenetic controller of MYPT1 to recruit with high temporal and spatial resolution protein phosphatase 1 to the cortex so that it can reduce the phosphorylation of myosin II light chain. By dephosphorylating MLC, myosin II is presumed to have lower activity in that region, leading to a reduction of myosin II-mediated force production. The authors then use their system to attempt to discern the myosin II contribution to tension in various locations around the cell, including during cytokinesis.

Overall, the strength of this paper is the potential of the opto-MYPT1 controller that the authors have developed. They do a reasonable job testing the effects of the controller. However, the major biological insights in this paper are fairly small given the history of deciphering the mechanics of cytokinesis. This is fine for this paper, as I would view this paper as more of a tool development study. Some questions do persist regarding what the authors are actually examining that come up throughout the paper.

Specific comments/questions:

Reviewer 3-Question 1 (R3-Q1): *1. The authors suggest that by recruiting PP1c that they are then specifically reducing phosphorylation of myosin II. But, PP1c appears to be fairly promiscuous and there are other known substrates such as moesin and adducin. What other substrates can their opto-MYPT1(169) control promote dephosphorylation of if all that is happening is PP1c is being recruited to the cortex, especially since the MLC binding site has been removed?*

Reviewer 3-Answer 1 (R3-A1): We share the reviewer’s concern. At this moment, we have not fully addressed the issue of specificity, but our data indicated, at least in part, that OptoMYPT exhibited specificity to the extent that it did not dephosphorylate ERM proteins (Fig. S1e). Further, we could not find other phospho-specific antibodies for the PP1c substrates. For example, adducin is regulated by phosphorylation at multiple sites, and phosphorylation sites that are dephosphorylated by PP1c have not been identified (Kiss, 2019, Biochimica et Biophysica Acta). Therefore, we have carefully

described the concerns in regard to specificity and the possible ways to improve these experiments in the Discussion of the revised manuscript as follow:

(page 19, line 529)

“There still remain some issues to be addressed with respect to the OptoMYPT. First is the issue of substrate specificity in OptoMYPT. We could not exclude the possibility that the OptoMYPT dephosphorylates additional substrates other than MLC. In addition, the expression of PP1BD of MYPT affected the localization of endogenous PP1c (Fig. 1d and Supplementary Fig. 1c). Some proteins are dephosphorylated by PP1c in the nucleus (Kiss et al. 2008), and therefore the nuclear exclusion of PP1c by OptoMYPT may disturb its nuclear function. However, based on the fact that the expression of MYPT296 did not dephosphorylate ERM proteins (Supplementary Fig. 1e), it seems somewhat unlikely that OptoMYPT dephosphorylates and inactivates the ERM proteins.”

R3-Q2: 2. *They do look at a pan-phospho-ERM in Fig. S1, but oddly this is never described in the Results section. I am not sure that pan-phospho-ERM is specific enough however. Two points here: this result needs to be pointed out in the Results, but with more specific probing should be provided if possible. What is the relative abundance of the different ERMs? Even if moesin is much less abundant, that does not mean that its impact is just as significant (relative abundance alone is not a sole indicator of contribution).*

R3-A2: We are sorry that our original description of the method was unclear. We included the following explanation in regard to ERM in the Result section of the revised manuscript:

(page 5, line 157)

“The ezrin-radixin-moesin (ERM) family of plasma membrane-actin cytoskeleton cross-linking proteins, which are known to be the substrates of PP1c, maintained their phosphorylation levels with pan-phospho-ERM antibody (Supplementary Fig. 1e), suggesting that the expression of PP1BD of MYPT has no impact on phosphorylation of the ERM proteins under these conditions.”

Regarding the relative abundance of ERM and phospho-ERM, we could not obtain phospho-specific antibodies for each of ezrin, radixin, and moesin. Further, as the reviewer mentioned, the relative abundance of ERMs is not a sole indicator of the contribution. Therefore, we did not examine the relative abundance of the different ERMs.

R3-Q3: 3. *Fig. 2C, given the nearly bimodal nature of the control group, a Student t-test would not be the appropriate statistical test. If one considered a non-normal distribution, are these data still considered statistically significant?*

R3-A3: As the reviewer pointed out, the assumptions of a Student's *t*-test might be too strict for the data. Instead, we adopted the Brunner-Munzel test, which allows both unequal variances and non-normality, and confirmed the statistical significance. We also obtained additional data and replaced the

original images with new ones in the revised manuscript (Fig. 2b, 2c), following the suggestion of Reviewer #1 (see **R1-Q5**).

R3-Q4: *4. Critical controls seem to be missing. It would be useful to see movies of the controls for the traction force experiment and the protrusion formation corresponding to movies S3 and S4, respectively. The effects provided are not overwhelming. Similar quantification of controls should be provided for Fig. 2f. The authors do provide an example image in Fig. 3 of control cells along with quantification, but again it would be useful to see example movies.*

R3-A4: According to the reviewer's suggestion, we have added a movie showing the control experiments for the analysis of membrane protrusion formation (Movie S2, upper) and the traction force experiment (Movie S6, left). In addition, we have included a representative image and data of the control experiment in the traction force analysis (Fig. 3b, 3c).

R3-Q5: *5. The premise about the controversy over the relative contributions of cortical tension to cytokinesis is extremely selective and overlooks considerable amount of research on this topic. This text needs to be significantly modernized.*

R3-A5: We agree with the reviewer. There was too much emphasis on the controversy in the original manuscript, particularly since this controversy has largely been resolved by recent studies. Thus, we carefully revised the relevant passage part to include recent reports as follows:

(page 13, line 349)

“We next applied the OptoMYPT system to elucidate the mechanical regulation of the actin cortex during cytokinesis. In this process, actin, NMII, and cross-linkers constitute a contractile ring in the equatorial plane, and generate force to divide a cell into two daughter cells by constriction (Pollard 2010; Green, Paluch, and Oegema 2012) (Fig. 4a, solid arrows). On the other hand, the tension developed in cortical actomyosin counteracts the force in the contractile ring (Yoneda and Dan 1972; Turlier et al. 2014; Stachowiak et al. 2014; Rodrigues et al. 2015; Kunda et al. 2008; Chapa-Y-Lazo et al. 2020) (Fig. 4a, dashed arrows). Thus, to advance the constriction, the contractile ring has to overcome the resistance of the actin cortex. There are two mechanisms for driving the cytokinesis; increasing tension in the ring and mechanical weakening of the cell cortex. The latter mechanism, so-called “polar relaxation”, through, for example, NMII removal from the polar region, is required for proper cytokinesis (Yoneda and Dan 1972; Turlier et al. 2014; Stachowiak et al. 2014; Rodrigues et al. 2015; Kunda et al. 2008; Chapa-Y-Lazo et al. 2020), but genetic and pharmacological inhibition of cortical actomyosin often induce cytokinetic failure (Taneja et al. 2020; Yamamoto et al. 2019; O’Connell, Warner, and Wang 2001; Wiggan et al. 2012). To elucidate the underlying mechanics, the strength of cortical tension has been directly measured using atomic force microscopy, microaspiration, and laser ablation methods (Taneja et al. 2020; Matzke, Jacobson, and Radmacher 2001; Tinevez et al. 2009). However, clarifying the contribution of cortical tension is still technically challenging due to the highly dynamic nature of cytokinesis.”

R3-Q6: 6. *Bleb formation results from the balance between cortical tension and membrane-cortex anchoring density so simply counting blebs as an indicator of cortical tension might be correct, but it might not be. Also, as mentioned above, other proteins, such as moesin, that help mediate membrane-cortex anchoring, are also reported as potential substrates of PP1c. Also, some of the anchoring proteins require myosin II to tug on them to get them to lock in. Further, in the Opto-MYPT1(169) scenario where you are still getting blebs, but bigger ones, this could be because the membrane-cortex linkages are even weaker. These effects are complicated for sure. Therefore, I think it is better to just be descriptive in what the changes in cell phenotype are, rather than jumping to the conclusion that cortical tension is all that is getting affected.*

R3-A6: We thank the reviewer for the nice suggestion. As the reviewer pointed out, the causal link between cortical tension and bleb count is a long way from straightforward, and our writing overly simplified it. Thus, we have substantially rewritten the relevant passage as follows:

(page 14, line 376)

“We focused on the dynamics of membrane blebbing at the polar cortex during cytokinesis, because bleb formation requires the intracellular pressure generated by cortical tension to be high enough to overcome membrane-cortex anchoring and surface tension of the plasma membrane (Sedzinski et al. 2011; Tinevez et al. 2009; Charras and Paluch 2008). In our experiments, the level of membrane bleb formation following the onset of cleavage furrow ingression was lower in OptoMYPT-pole cells than Control-pole or OptoMYPT-dark cells (Fig. 4c, d). In addition, we found that the local activation of OptoMYPT altered the onset and size of membrane bleb formation during cytokinesis (Fig. 4e, Supplementary Figure 7). First, the onset of blebbing was delayed in OptoMYPT-pole cells, while Control-pole and OptoMYPT-dark cells showed blebbing from the early phase of ring constriction. Second, in the later phase of cytokinesis, OptoMYPT-pole cells exhibited large blebs (Supplementary Fig. 7), although the bleb counts were still smaller than the control cases. We should note that the OptoMYPT activation might weaken the membrane-cortex linkage, such as by ERM deactivation. However, this weakening of the membrane-cortex linkage has been shown to render the OptoMYPT-pole cells more prone to bleb formation (Rodrigues et al. 2015; Charras and Paluch 2008). Therefore, the absence of blebbing in the early phase indicates reduced tension in OptoMYPT-pole cells independent of whether membrane-cortex linkage is weakened or not. On the other hand, the larger blebs of the OptoMYPT-pole cells in the late phase, possibly initiated by excess intracellular pressure coming from ring constriction, might be explained by both the weakening of the membrane-cortex linkage and inefficient bleb retraction (see the Discussion section for more details).”

R3-Q7: 7. *I agree that changes in the rates of furrow ingression can reflect differences cortical tension, but myosin II can also impact strain stiffening of the network, which similarly can impact furrow ingression rates. In fact, both effects are contributors to furrow ingression rates. I would simplify this portion of the discussion without solely implicating one parameter vs. the other – better to be inclusive of potential mechanisms.*

R3-A7: According to the reviewer's suggestion, we revised the corresponding part to consider both the active (i.e., myosin II) and passive (i.e., viscoelasticity of the cortices) contributions to the cortical tension as follows:

(page 15, line 406)

"The cortical tension represents both the myosin-generated tension and the viscoelastic response of the cortical cytoskeleton. We note that the OptoMYPT activation might have modulated not only the myosin-generated tension but also the viscoelastic response through, for example, inhibition of the cross-linking activity of NMII."

We also modified our explanation of our model in order to emphasize that the cortical tension in the model equation can come not only from myosin-generated forces but also from viscoelastic responses of the cortical cytoskeletal network as follows:

(page 2, line 38 in Supplementary Information)

"Thus, the cortical tension depends on both the mechanical strength of the cortical cytoskeleton, which causes the effective viscosity, and myosin motors, which cause active tension. Here we simply describe the cortices as an active viscous membrane having tension T_c ."

R3-Q8: It should be appreciated that the authors do point out many of these issues described above in the Discussion, which I was happy to see. I think they need to go back through Results and present each point similarly using qualified language so that they do not over-interpret their observations from the outset.

R3-A8: We would like to thank the reviewer for this remark. We have carefully revised the Results section to avoid over-interpretation of our findings.

Reviewers' Comments:

Reviewer #1:

Remarks to the Author:

The authors have satisfactorily answered most of the the concerns and questions that were raised earlier, and overall the manuscript has improved. Extra evidence has now been provided that convincingly demonstrate a reduction in contractility near the cell cortex upon blue light exposure. Especially, the extra experiments performed in *Xenopus* embryos, looking at cell junction morphology are a nice addition. (I did find it surprising that this exp did not work in MDCK cells).

The difference in phosphorylation of MLC upon blue light stimulation is really quite small (on westernblot suppl fig 4a,b), but I understand that the authors might only be targeting a small population near the PM...

The authors now properly discuss many of the caveats of the approach/tool in the discussion section, although I think that the authors could immediately point out possible specificity issues in the results section. Now they indirectly do this by saying that there's no impact on ERM phosphorylation, which is of course a good thing. But I suggest that the authors should first write a sentence about possible specificity issues, but that at least for ERM family this does not seem to be the case.

Although I could argue that not sufficient evidence is provided for that, because the authors did not look at ERM phosphorylation after blue light. It is not unlikely that recruitment of PP1C to the plasma membrane could impact ERM phosphorylation just as well as it impacts MLC phosphorylation, since these proteins are concentrated near the plasma membrane. Especially since it appears that all parts of opto-MYPT that make it specific for myosin have been removed, and it still can dephosphorylate MLC near the cortex...

A small final suggestion:

In the schematic in figure 1c, the authors should add the NES sequence, to make it clear from the start that this sequence was added.

With minor modifications I now support the publication of this manuscript in Nature Communications.

Reviewer #2:

Remarks to the Author:

Review report Aoki, 2021 Nat Com. (Revision)

Yamamoto and colleagues have revised the manuscript considerably, and addressed most my comments satisfactory.

Looking at the entire discussion and comments of the other reviewers, I still have some doubt regarding the small effect that the authors describe and the specificity of the probe to deactivate Myosin II. The specificity is hard to address without good antibodies for the other candidates of phosphorylation by PP1c, and the authors sufficiently describe the limitations in the text now.

However, the lack of effect on actin dynamics/stress fibre content in the cell is surprising. The authors address this issue by stating that the effect of the tool is limited to the pool of Myosin II that is in the vicinity of the plasma membrane. This might be true. A simple TIRF experiment (as suggested by reviewer 1), can provide definitive proof that this tool can be used to dephosphorylate Myosin II bound to actin bundles. Showing this would greatly improve the strength of this manuscript, and the usability of the tool in my opinion.

Reviewer #3:

Remarks to the Author:

The authors have adequately addressed my questions in their revision. I think this paper can now move forward, assuming the other reviewers are also satisfied.

General Statements

We thank the reviewers for their critical comments and valuable suggestions, which have helped us to improve our paper. Our point-by-point responses are given below, with the comments of the reviewers shown in *dark blue italics*. As detailed in our responses, we revised and expanded numerous explanatory passages in the main text.

=====

REVIEWER COMMENTS

Reviewer #1 (Remarks to the Author):

The authors have satisfactorily answered most of the the concerns and questions that were raised earlier, and overall the manuscript has improved. Extra evidence has now been provided that convincingly demonstrate a reduction in contractility near the cell cortex upon blue light exposure. Especially, the extra experiments performed in Xenopus embryos, looking at cell junction morphology are a nice addition. (I did find it surprising that this exp did not work in MDCK cells).

The difference in phosphorylation of MLC upon blue light stimulation is really quite small (on westernblot suppl fig 4a,b), but I understand that the authors might only be targeting a small population near the PM...

The authors now properly discuss many of the caveats of the approach/tool in the discussion section, although I think that the authors could immediately point out possible specificity issues in the results section. Now they indirectly do this by saying that there's no impact on ERM phosphorylation, which is of course a good thing. But I suggest that the authors should first write a sentence about possible specificity issues, but that at least for ERM family this does not seem to be the case. Although I could argue that not sufficient evidence is provided for that, because the authors did not look at ERM phosphorylation after blue light. It is not unlikely that recruitment of PP1C to the plasma membrane could impact ERM phosphorylation just as well as it impacts MLC phosphorylation, since these proteins are concentrated near the plasma membrane. Especially since it appears that all parts of opto-MYPT that make it specific for myosin have been removed, and it still can dephosphorylate MLC near the cortex...

We would like to thank the reviewer for carefully reading the manuscript and the valuable comments. According to the reviewer's suggestion, we have included the following sentence to mention the specificity issue of OptoMYPT in the Results section of the revised manuscript:

(Page 4, line 114)

"Since PP1c is known to dephosphorylate various substrates, it should be kept in mind that the OptoMYPT dephosphorylates non-MLC substrates such as ezrin-radixin-moesin (ERM) family of plasma membrane-actin cytoskeleton cross-linking proteins."

A small final suggestion:

In the schematic in figure 1c, the authors should add the NES sequence, to make it clear from the start that this sequence was added.

We agree with the reviewer's suggestion, and we have included the NES sequence in the Figure 1c and Supplementary Figure 2a.

=====

Reviewer #2 (Remarks to the Author):

Reviewer #2 (Remarks to the Author):

Review report Aoki, 2021 Nat Com. (Revision)

Yamamoto and colleagues have revised the manuscript considerably, and addressed most my comments satisfactory.

Looking at the entire discussion and comments of the other reviewers, I still have some doubt regarding the small effect that the authors describe and the specificity of the probe to deactivate Myosin II. The specificity is hard to address without good antibodies for the other candidates of phosphorylation by PP1c, and the authors sufficiently describe the limitations in the text now.

However, the lack of effect on actin dynamics/stress fibre content in the cell is surprising. The authors address this issue by stating that the effect of the tool is limited to the pool of Myosin II that is in the vicinity of the plasma membrane. This might be true. A simple TIRF experiment (as suggested by reviewer 1), can provide definitive proof that this tool can be used to dephosphorylate Myosin II bound to actin bundles. Showing this would greatly improve the strength of this manuscript, and the usability of the tool in my opinion.

We would like to thank the reviewer for further evaluating our work. As suggested, we tried to visualize ppMLC signals at the plasma membrane by a TIRF microscope with collaborators. However, we recognized that it was technically difficult to quantitatively assess the ppMLC levels by TIRF, because of the high sensitivity of the reflection angle of excitation lasers to the fluorescence signals. Therefore, we gave up the use of TIRF microscopy. Next, we used a conventional confocal laser scanning microscope (Leica SP8) with a high NA objective lens (x100, NA1.4) and a short wavelength (405 nm) excitation laser to improve spatial resolutions. Under this condition, we took images of ppMLC stained with anti-ppMLC antibodies and Alexa-405 secondary antibodies at the bottom plasma membrane. As shown below, there are puncta signals (upper), which are typical ppMLC patterns (A.M. Fenix et al., *Mol. Biol. Cell* (2016), J.R. Beach et al., *Nat. Cell Biol.* (2017), S. Hu et al., *Nat. Cell Biol.* (2017)). Of note, these puncta were also observed by using the above TIRF microscope. In addition, ppMLC signals appeared to be reduced at the protrusive area in OptoMYPT-expressing cells with blue light illumination (right). However, due to the high heterogeneity of cell morphology and ppMLC staining pattern, careful image acquisition and image analysis are necessary to demonstrate the decrease in

ppMLC at the plasma membranes quantitatively. Thus, we have not included these data in the revised manuscript.

Reviewer only figure legend. Confocal imaging of phosphorylated MLC at the lamellipodial region.

Immunofluorescence analysis of phosphorylated MLC in MDCK cells transiently expressing SspB-mScarlet-I (left) and SspB-mScarlet-I-MYPT169 (middle and right) with Stargazin-mEGFP-iLID. In Control-light and OptoMYPT-light conditions, blue light was illuminated from the top of the stage for 30 min. After illumination, cells were immediately fixed, and the subsequent staining was processed according to the method described in the main manuscript. Representative 2 cells for each condition are shown. Scale bar, 5 μ m.

Reviewer #3 (Remarks to the Author):

The authors have adequately addressed my questions in their revision. I think this paper can now move forward, assuming the other reviewers are also satisfied.

We would like to thank the reviewer for the support of our study.